# A primal-dual method for conic constrained distributed optimization problems

**Necdet Serhat Aybat**
Department of Industrial Engineering
Penn State University
University Park, PA 16802
nsa10@psu.edu

**Erfan Yazdandoost Hamedani**
Department of Industrial Engineering
Penn State University
University Park, PA 16802
evy5047@psu.edu

## Abstract

We consider cooperative multi-agent consensus optimization problems over an undirected network of agents, where only those agents connected by an edge can directly communicate. The objective is to minimize the sum of agent-specific composite convex functions over agent-specific private conic constraint sets; hence, the optimal consensus decision should lie in the intersection of these private sets. We provide convergence rates in sub-optimality, infeasibility and consensus violation; examine the effect of underlying network topology on the convergence rates of the proposed decentralized algorithms; and show how to extend these methods to handle time-varying communication networks.

## 1 Introduction

Let $\mathcal{G} = (\mathcal{N}, \mathcal{E})$ denote a *connected* undirected graph of $N$ computing nodes, where $\mathcal{N} \triangleq \{1, \ldots, N\}$ and $\mathcal{E} \subseteq \mathcal{N} \times \mathcal{N}$ denotes the set of edges – without loss of generality assume that $(i, j) \in \mathcal{E}$ implies $i < j$. Suppose nodes $i$ and $j$ can exchange information only if $(i, j) \in \mathcal{E}$, and each node $i \in \mathcal{N}$ has a *private* (local) cost function $\Phi_i : \mathbb{R}^n \to \mathbb{R} \cup \{+\infty\}$ such that

$$\Phi_i(x) \triangleq \rho_i(x) + f_i(x), \tag{1}$$

where $\rho_i : \mathbb{R}^n \to \mathbb{R} \cup \{+\infty\}$ is a possibly *non-smooth* convex function, and $f_i : \mathbb{R}^n \to \mathbb{R}$ is a *smooth* convex function. We assume that $f_i$ is differentiable on an open set containing $\mathbf{dom}\,\rho_i$ with a Lipschitz continuous gradient $\nabla f_i$, of which Lipschitz constant is $L_i$; and the prox map of $\rho_i$,

$$\mathbf{prox}_{\rho_i}(x) \triangleq \operatorname*{argmin}_{y \in \mathbb{R}^n} \left\{ \rho_i(y) + \tfrac{1}{2} \|y - x\|^2 \right\}, \tag{2}$$

is *efficiently* computable for $i \in \mathcal{N}$, where $\|.\|$ denotes the Euclidean norm. Let $\mathcal{N}_i \triangleq \{j \in \mathcal{N} : (i, j) \in \mathcal{E} \text{ or } (j, i) \in \mathcal{E}\}$ denote the set of neighboring nodes of $i \in \mathcal{N}$, and $d_i \triangleq |\mathcal{N}_i|$ is the degree of node $i \in \mathcal{N}$. Consider the following minimization problem:

$$\min_{x \in \mathbb{R}^n} \sum_{i \in \mathcal{N}} \Phi_i(x) \quad \text{s.t.} \quad A_i x - b_i \in \mathcal{K}_i, \quad \forall i \in \mathcal{N}, \tag{3}$$

where $A_i \in \mathbb{R}^{m_i \times n}$, $b_i \in \mathbb{R}^{m_i}$ and $\mathcal{K}_i \subseteq R^{m_i}$ is a closed, convex cone. Suppose that projections onto $\mathcal{K}_i$ can be computed efficiently, while the projection onto the *preimage* $A_i^{-1}(\mathcal{K}_i + b_i)$ is assumed to be *impractical*, e.g., when $\mathcal{K}_i$ is the positive semidefinite cone, projection to preimage requires solving an SDP. Our objective is to solve (3) in a decentralized fashion using the computing nodes $\mathcal{N}$ and exchanging information only along the edges $\mathcal{E}$. In Section 2 and Section 3, we consider (3) when the topology of the connectivity graph is *static* and *time-varying*, respectively.

This computational setting, i.e., decentralized consensus optimization, appears as a generic model for various applications in signal processing, e.g., [1, 2], machine learning, e.g., [3, 4, 5] and statistical inference, e.g., [6]. Clearly, (3) can also be solved in a "centralized" fashion by communicating all the private functions $\Phi_i$ to a *central* node, and solving the overall problem at this

node. However, such an approach can be very expensive both from communication and computation perspectives when compared to the distributed algorithms which are far more scalable to increasing problem data and network sizes. In particular, suppose $(A_i, b_i) \in \mathbb{R}^{m \times (n+1)}$ and $\Phi_i(x) = \lambda \|x\|_1 + \|A_i x - b_i\|^2$ for some given $\lambda > 0$ for $i \in \mathcal{N}$ such that $m \ll n$ and $N \gg 1$. Hence, (3) is a very large scale LASSO problem with *distributed* data. To solve (3) in a centralized fashion, the data $\{(A_i, b_i) : i \in \mathcal{N}\}$ needs to be communicated to the central node. This can be prohibitively expensive, and may also violate privacy constraints – in case some node $i$ does not want to reveal the details of its *private* data. Furthermore, it requires that the central node has large enough memory to be able to accommodate all the data. On the other hand, at the expense of slower convergence, one can completely do away with a central node, and seek for *consensus* among all the nodes on an optimal decision using "local" decisions communicated by the neighboring nodes. From computational perspective, for certain cases, computing partial gradients *locally* can be more computationally efficient when compared to computing the entire gradient at a central node. With these considerations in mind, we propose decentralized algorithms that can compute solutions to (3) using only *local* computations without explicitly requiring the nodes to communicate the functions $\{\Phi_i : i \in \mathcal{N}\}$; thereby, circumventing all privacy, communication and memory issues. Examples of *constrained* machine learning problems that fit into our framework include multiple kernel learning [7], and primal linear support vector machine (SVM) problems. In the numerical section we implemented the proposed algorithms on the primal SVM problem.

## 1.1 Previous Work

There has been active research [8, 9, 10, 11, 12] on solving convex-concave saddle point problems $\min_x \max_y \mathcal{L}(x, y)$. In [9] primal-dual proximal algorithms are proposed for convex-concave problems with known saddle-point structure $\min_x \max_y \mathcal{L}_s(x, y) \triangleq \Phi(x) + \langle Tx, y \rangle - h(y)$, where $\Phi$ and $h$ are convex functions, and $T$ is a linear map. These algorithms converge with rate $\mathcal{O}(1/k)$ for the primal-dual gap, and they can be modified to yield a convergence rate of $\mathcal{O}(1/k^2)$ when either $\Phi$ or $h$ is strongly convex, and $\mathcal{O}(1/e^k)$ linear rate, when both $\Phi$ and $h$ are strongly convex. More recently, in [11] Chambolle and Pock extend their previous work in [9], using simpler proofs, to handle composite convex primal functions, i.e., sum of smooth and (possibly) nonsmooth functions, and to deal with proximity operators based on Bregman distance functions.

Consider $\min_{x \in \mathbb{R}^n} \{\sum_{i \in \mathcal{N}} \Phi_i(x) : x \in \cap_{i \in \mathcal{N}} \mathcal{X}_i\}$ over $\mathcal{G} = (\mathcal{N}, \mathcal{E})$. Although the *unconstrained* consensus optimization, i.e., $\mathcal{X}_i = \mathbb{R}^n$, is well studied – see [13, 14] and the references therein, the *constrained* case is still an immature, and recently developing area of active research [13, 14, 15, 16, 17, 18, 19]. Other than few exceptions, e.g., [15, 16, 17], the methods in literature require that each node compute a projection on the privately known set $\mathcal{X}_i$ in addition to consensus and (sub)gradient steps, e.g., [18, 19]. Moreover, among those few exceptions that do not use projections onto $\mathcal{X}_i$ when $\Pi_{\mathcal{X}_i}$ is not easy to compute, only [15, 16] can handle agent-specific constraints without assuming global knowledge of the constraints by all agents. However, *no* rate results in terms of suboptimality, local infeasibility, and consensus violation exist for the primal-dual distributed methods in [15, 16] when implemented for the agent-specific conic constraint sets $\mathcal{X}_i = \{x : A_i x - b_i \in \mathcal{K}_i\}$ studied in this paper. In [15], a consensus-based distributed primal-dual perturbation (PDP) algorithm using a square summable but not summable step-size sequence is proposed. The objective is to minimize a composition of a global network function (smooth) with the summation of local objective functions (smooth), subject to local compact sets and inequality constraints on the summation of agent specific constrained functions. They showed that the local primal-dual iterate sequence converges to a global optimal primal-dual solution; however, no rate result was provided. The proposed PDP method can also handle non-smooth constraints with similar convergence guarantees. Finally, while we were preparing this paper, we became aware of a very recent work [16] related to ours. The authors proposed a distributed algorithm on time-varying communication network for solving saddle-point problems subject to consensus constraints. The algorithm can also be applied to solve consensus optimization problems with inequality constraints that can be written as summation of local convex functions of local and global variables. Under some assumptions, it is shown that using a carefully selected decreasing step-size sequence, the ergodic average of primal-dual sequence converges with $\mathcal{O}(1/\sqrt{k})$ rate in terms of saddle-point evaluation error; however, when applied to constrained optimization problems, *no* rate in terms of either suboptimality or infeasibility is provided.

**Contribution.** We propose primal-dual algorithms for distributed optimization subject to agent specific conic constraints. By assuming composite convex structure on the primal functions, we show that our proposed algorithms converge with $\mathcal{O}(1/k)$ rate where $k$ is the number of consensus iterations. To the best of our knowledge, this is the best rate result for our setting. Indeed, $\epsilon$-optimal and $\epsilon$-feasible solution can be computed within $\mathcal{O}(1/\epsilon)$ consensus iterations for the static topology, and within $\mathcal{O}(1/\epsilon^{1+1/p})$ consensus iterations for the dynamic topology for any rational $p \geq 1$, although $\mathcal{O}(1)$ constant gets larger for large $p$. Moreover, these methods are fully distributed, i.e., the agents are *not* required to know any global parameter depending on the entire network topology, e.g., the second smallest eigenvalue of the Laplacian; instead, we only assume that agents know who their neighbors are. Due to limited space, we put all the technical proofs to the appendix.

## 1.2 Preliminary

Let $\mathcal{X}$ and $\mathcal{Y}$ be finite-dimensional vector spaces. In a recent paper, Chambolle and Pock [11] proposed a primal-dual algorithm (PDA) for the following convex-concave saddle-point problem:

$$\min_{\mathbf{x}\in\mathcal{X}} \max_{\mathbf{y}\in\mathcal{Y}} \mathcal{L}(\mathbf{x},\mathbf{y}) \triangleq \Phi(\mathbf{x}) + \langle T\mathbf{x},\mathbf{y}\rangle - h(\mathbf{y}), \quad \text{where} \quad \Phi(\mathbf{x}) \triangleq \rho(\mathbf{x}) + f(\mathbf{x}), \tag{4}$$

$\rho$ and $h$ are possibly non-smooth convex functions, $f$ is a convex function and has a Lipschitz continuous gradient defined on $\mathbf{dom}\,\rho$ with constant $L$, and $T$ is a linear map. Briefly, given $\mathbf{x}^0, \mathbf{y}^0$ and algorithm parameters $\nu_x, \nu_y > 0$, PDA consists of two proximal-gradient steps:

$$\mathbf{x}^{k+1} \leftarrow \operatorname*{argmin}_{\mathbf{x}} \rho(\mathbf{x}) + f(\mathbf{x}^k) + \left\langle \nabla f(\mathbf{x}^k),\, \mathbf{x} - \mathbf{x}^k \right\rangle + \left\langle T\mathbf{x}, \mathbf{y}^k \right\rangle + \frac{1}{\nu_x} D_x(\mathbf{x}, \mathbf{x}^k) \tag{5a}$$

$$\mathbf{y}^{k+1} \leftarrow \operatorname*{argmin}_{\mathbf{y}} h(\mathbf{y}) - \left\langle T(2\mathbf{x}^{k+1} - \mathbf{x}^k), \mathbf{y} \right\rangle + \frac{1}{\nu_y} D_y(\mathbf{y}, \mathbf{y}^k), \tag{5b}$$

where $D_x$ and $D_y$ are Bregman distance functions corresponding to some continuously differentiable strongly convex $\psi_x$ and $\psi_y$ such that $\mathbf{dom}\,\psi_x \supset \mathbf{dom}\,\rho$ and $\mathbf{dom}\,\psi_y \supset \mathbf{dom}\,h$. In particular, $D_x(\mathbf{x},\bar{\mathbf{x}}) \triangleq \psi_x(\mathbf{x}) - \psi_x(\bar{\mathbf{x}}) - \langle \nabla\psi_x(\bar{\mathbf{x}}),\, \mathbf{x} - \bar{\mathbf{x}} \rangle$, and $D_y$ is defined similarly. In [11], a simple proof for the ergodic convergence is provided for (5); indeed, it is shown that, when the convexity modulus for $\psi_x$ and $\psi_y$ is 1, if $\tau, \kappa > 0$ are chosen such that $(\frac{1}{\nu_x} - L)\frac{1}{\nu_y} \geq \sigma_{\max}^2(T)$, then

$$\mathcal{L}(\bar{\mathbf{x}}^K, \mathbf{y}) - \mathcal{L}(\mathbf{x}, \bar{\mathbf{y}}^K) \leq \frac{1}{K}\left( \frac{1}{\nu_x} D_x(\mathbf{x}, \mathbf{x}^0) + \frac{1}{\nu_y} D_y(\mathbf{y}, \mathbf{y}^0) - \left\langle T(\mathbf{x} - \mathbf{x}^0), \mathbf{y} - \mathbf{y}^0 \right\rangle \right), \tag{6}$$

for all $\mathbf{x}, \mathbf{y} \in \mathcal{X} \times \mathcal{Y}$, where $\bar{\mathbf{x}}^K \triangleq \frac{1}{K}\sum_{k=1}^K \mathbf{x}^k$ and $\bar{\mathbf{y}}^K \triangleq \frac{1}{K}\sum_{k=1}^K \mathbf{y}^k$.

First, we define the notation used throughout the paper. Next, in Theorem 1.1, we discuss a special case of (4), which will help us prove the main results of this paper, and also allow us to develop decentralized algorithms for the consensus optimization problem in (3). The proposed algorithms in this paper can distribute the computation over the nodes such that each node's computation is based on the local topology of $\mathcal{G}$ and the private information only available to that node.

**Notation.** Throughout the paper, $\|.\|$ denotes the Euclidean norm. Given a convex set $\mathcal{S}$, let $\sigma_{\mathcal{S}}(.)$ denote its support function, i.e., $\sigma_{\mathcal{S}}(\theta) \triangleq \sup_{w\in\mathcal{S}} \langle \theta,\, w \rangle$, let $\mathbb{I}_{S}(\cdot)$ denote the indicator function of $\mathcal{S}$, i.e., $\mathbb{I}_S(w) = 0$ for $w \in \mathcal{S}$ and equal to $+\infty$ otherwise, and let $\mathcal{P}_S(w) \triangleq \operatorname{argmin}\{\|v - w\| : v \in \mathcal{S}\}$ denote the projection onto $\mathcal{S}$. For a closed convex set $\mathcal{S}$, we define the distance function as $d_{\mathcal{S}}(w) \triangleq \|\mathcal{P}_S(w) - w\|$. Given a convex cone $\mathcal{K} \in \mathbb{R}^m$, let $\mathcal{K}^*$ denote its dual cone, i.e., $\mathcal{K}^* \triangleq \{\theta \in \mathbb{R}^m : \langle \theta, w \rangle \geq 0 \ \forall w \in \mathcal{K}\}$, and $\mathcal{K}^{\circ} \triangleq -\mathcal{K}^*$ denote the polar cone of $\mathcal{K}$. Note that for a given cone $\mathcal{K} \in \mathbb{R}^m$, $\sigma_{\mathcal{K}}(\theta) = 0$ for $\theta \in \mathcal{K}^{\circ}$ and equal to $+\infty$ if $\theta \notin \mathcal{K}^{\circ}$, i.e., $\sigma_{\mathcal{K}}(\theta) = \mathbb{I}_{\mathcal{K}^{\circ}}(\theta)$ for all $\theta \in \mathbb{R}^m$. Cone $\mathcal{K}$ is called *proper* if it is closed, convex, pointed, and it has a nonempty interior. Given a convex function $g : \mathbb{R}^n \to \mathbb{R} \cup \{+\infty\}$, its convex conjugate is defined as $g^*(w) \triangleq \sup_{\theta\in\mathbb{R}^n} \langle w, \theta \rangle - g(\theta)$. $\otimes$ denotes the Kronecker product, and $\mathbf{I}_n$ is the $n \times n$ identity matrix.

**Definition 1.** *Let $\mathcal{X} \triangleq \Pi_{i\in\mathcal{N}}\mathbb{R}^n$ and $\mathcal{X} \ni \mathbf{x} = [x_i]_{i\in\mathcal{N}}$; $\mathcal{Y} \triangleq \Pi_{i\in\mathcal{N}}\mathbb{R}^{m_i} \times \mathbb{R}^{m_0}$, $\mathcal{Y} \ni \mathbf{y} = [\boldsymbol{\theta}^\top \boldsymbol{\lambda}^\top]^\top$ and $\boldsymbol{\theta} = [\theta_i]_{i\in\mathcal{N}} \in \mathbb{R}^m$, where $m \triangleq \sum_{i\in\mathcal{N}} m_i$, and $\Pi$ denotes the Cartesian product. Given parameters $\gamma > 0$, $\kappa_i, \tau_i > 0$ for $i \in \mathcal{N}$, let $\mathbf{D}_\gamma \triangleq \frac{1}{\gamma}\mathbf{I}_{m_0}$, $\mathbf{D}_\kappa \triangleq \mathbf{diag}([\frac{1}{\kappa_i}\mathbf{I}_{m_i}]_{i\in\mathcal{N}})$, and $\mathbf{D}_\tau \triangleq \mathbf{diag}([\frac{1}{\tau_i}\mathbf{I}_n]_{i\in\mathcal{N}})$. Defining $\psi_x(\mathbf{x}) \triangleq \frac{1}{2}\mathbf{x}^\top \mathbf{D}_\tau \mathbf{x}$ and $\psi_y(\mathbf{y}) \triangleq \frac{1}{2}\boldsymbol{\theta}^\top \mathbf{D}_\kappa \boldsymbol{\theta} + \frac{1}{2}\boldsymbol{\lambda}^\top \mathbf{D}_\gamma \boldsymbol{\lambda}$ leads to the following Bregman distance functions: $D_x(\mathbf{x},\bar{\mathbf{x}}) = \frac{1}{2}\|\mathbf{x} - \bar{\mathbf{x}}\|_{\mathbf{D}_\tau}^2$, and $D_y(\mathbf{y},\bar{\mathbf{y}}) = \frac{1}{2}\|\boldsymbol{\theta} - \bar{\boldsymbol{\theta}}\|_{\mathbf{D}_\kappa}^2 + \frac{1}{2}\|\boldsymbol{\lambda} - \bar{\boldsymbol{\lambda}}\|_{\mathbf{D}_\gamma}^2$, where the Q-norm is defined as $\|z\|_Q \triangleq (z^\top Q z)^{\frac{1}{2}}$ for $Q \succ 0$.*

**Theorem 1.1.** *Let $\mathcal{X}$, $\mathcal{Y}$, and Bregman functions $D_x$, $D_y$ be defined as in Definition 1. Suppose $\Phi(\mathbf{x}) \triangleq \sum_{i\in\mathcal{N}} \Phi_i(x_i)$, and $h(\mathbf{y}) \triangleq h_0(\boldsymbol{\lambda}) + \sum_{i\in\mathcal{N}} h_i(\theta_i)$, where $\{\Phi_i\}_{i\in\mathcal{N}}$ are composite convex functions defined as in* (1), *and $\{h_i\}_{i\in\mathcal{N}}$ are closed convex with simple prox-maps. Given $A_0 \in \mathbb{R}^{m_0\times n|\mathcal{N}|}$ and $\{A_i\}_{i\in\mathcal{N}}$ such that $A_i \in \mathbb{R}^{m_i\times n}$, let $T = [A^\top \ A_0^\top]^\top$, where $A \triangleq \mathbf{diag}([A_i]_{i\in\mathcal{N}}) \in \mathbb{R}^{m\times n|\mathcal{N}|}$ is a block-diagonal matrix. Given the initial point $(\mathbf{x}^0, \mathbf{y}^0)$, the PDA iterate sequence $\{\mathbf{x}^k, \mathbf{y}^k\}_{k\geq 1}$, generated according to* (5a) *and* (5b) *when $\nu_x = \nu_y = 1$, satisfies* (6) *for all $K \geq 1$ if $\bar{\mathbf{Q}}(A, A_0) \triangleq \begin{bmatrix} \bar{\mathbf{D}}_\tau & -A^\top & -A_0^\top \\ -A & \mathbf{D}_\kappa & 0 \\ -A_0 & 0 & \mathbf{D}_\gamma \end{bmatrix} \succeq 0$, where $\bar{\mathbf{D}}_\tau \triangleq \mathbf{diag}([(\frac{1}{\tau_i} - L_i)\mathbf{I}_n]_{i\in\mathcal{N}})$. Moreover, if a saddle point exists for* (4), *and $\bar{\mathbf{Q}}(A, A_0) \succ 0$, then $\{\mathbf{x}^k, \mathbf{y}^k\}_{k\geq 1}$ converges to a saddle point of* (4); *hence, $\{\bar{\mathbf{x}}^k, \bar{\mathbf{y}}^k\}_{k\geq 1}$ converges to the same point.*

Although the proof of Theorem 1.1 follows from the lines of [11], we provide the proof in the appendix for the sake of completeness as it will be used repeatedly to derive our results.

Next we discuss how (5) can be implemented to compute an $\epsilon$-optimal solution to (3) in a distributed way using only $\mathcal{O}(1/\epsilon)$ communications over the communication graph $\mathcal{G}$ while respecting node-specific privacy requirements. Later, in Section 3, we consider the scenario where the topology of the connectivity graph is time-varying, and propose a distributed algorithm that requires $\mathcal{O}(1/\epsilon^{1+\frac{1}{p}})$ communications for any $p \geq 1$. Finally, in Section 4 we test the proposed algorithms for solving the primal SVM problem in a decentralized manner. These results are shown under Assumption 1.1.

**Assumption 1.1.** *The duality gap for* (3) *is zero, and a primal-dual solution to* (3) *exists.*

A sufficient condition for this is the existence of a Slater point, i.e., there exists $\bar{x} \in \mathbf{relint}(\mathbf{dom}\,\Phi)$ such that $A_i\bar{x} - b_i \in \mathbf{int}(\mathcal{K}_i)$ for $i \in \mathcal{N}$, where $\mathbf{dom}\,\Phi = \cap_{i\in\mathcal{N}} \mathbf{dom}\,\Phi_i$.

## 2 Static Network Topology

Let $x_i \in \mathbb{R}^n$ denote the *local* decision vector of node $i \in \mathcal{N}$. By taking advantage of the fact that $\mathcal{G}$ is *connected*, we can reformulate (3) as the following *distributed consensus* optimization problem:

$$\min_{x_i\in\mathbb{R}^n,\, i\in\mathcal{N}} \left\{ \sum_{i\in\mathcal{N}} \Phi_i(x_i) \mid x_i = x_j : \lambda_{ij}, \ \forall (i,j)\in\mathcal{E}, \quad A_ix_i - b_i \in \mathcal{K}_i : \theta_i, \ \forall i\in\mathcal{N} \right\}, \quad (7)$$

where $\lambda_{ij} \in \mathbb{R}^n$ and $\theta_i \in \mathbb{R}^{m_i}$ are the corresponding dual variables. Let $\mathbf{x} = [x_i]_{i\in\mathcal{N}} \in \mathbb{R}^{n|\mathcal{N}|}$. The consensus constraints $x_i = x_j$ for $(i,j)\in\mathcal{E}$ can be formulated as $M\mathbf{x} = 0$, where $M \in \mathbb{R}^{n|\mathcal{E}|\times n|\mathcal{N}|}$ is a block matrix such that $M = H \otimes \mathbf{I}_n$ where $H$ is the oriented edge-node incidence matrix, i.e., the entry $H_{(i,j),l}$, corresponding to edge $(i,j)\in\mathcal{E}$ and $l \in \mathcal{N}$, is equal to 1 if $l = i$, $-1$ if $l = j$, and 0 otherwise. Note that $M^\top M = H^\top H \otimes \mathbf{I}_n = \Omega \otimes \mathbf{I}_n$, where $\Omega \in \mathbb{R}^{|\mathcal{N}|\times|\mathcal{N}|}$ denotes the graph Laplacian of $\mathcal{G}$, i.e., $\Omega_{ii} = d_i$, $\Omega_{ij} = -1$ if $(i,j)\in\mathcal{E}$ or $(j,i)\in\mathcal{E}$, and equal to 0 otherwise.

For any closed convex set $\mathcal{S}$, we have $\sigma_\mathcal{S}^*(\cdot) = \mathbb{I}_S(\cdot)$; therefore, using the fact that $\sigma_{\mathcal{K}_i}^* = \mathbb{I}_{\mathcal{K}_i}$ for $i \in \mathcal{N}$, one can obtain the following saddle point problem corresponding to (7),

$$\min_{\mathbf{x}} \max_{\mathbf{y}} \mathcal{L}(\mathbf{x}, \mathbf{y}) \triangleq \sum_{i\in\mathcal{N}} \left( \Phi_i(x_i) + \langle \theta_i, A_ix_i - b_i \rangle - \sigma_{\mathcal{K}_i}(\theta_i) \right) + \langle \boldsymbol{\lambda}, M\mathbf{x} \rangle, \quad (8)$$

where $\mathbf{y} = [\boldsymbol{\theta}^\top \ \boldsymbol{\lambda}^\top]^\top$ for $\boldsymbol{\lambda} = [\lambda_{ij}]_{(i,j)\in\mathcal{E}} \in \mathbb{R}^{n|\mathcal{E}|}$, $\boldsymbol{\theta} = [\theta_i]_{i\in\mathcal{N}} \in \mathbb{R}^m$, and $m \triangleq \sum_{i\in\mathcal{N}} m_i$.

Next, we study the distributed implementation of PDA in (5a)-(5b) to solve (8). Let $\Phi(\mathbf{x}) \triangleq \sum_{i\in\mathcal{N}} \Phi_i(x_i)$, and $h(\mathbf{y}) \triangleq \sum_{i\in\mathcal{N}} \sigma_{\mathcal{K}_i}(\theta_i) + \langle b_i, \theta_i \rangle$. Define the block-diagonal matrix $A \triangleq \mathbf{diag}([A_i]_{i\in\mathcal{N}}) \in \mathbb{R}^{m\times n|\mathcal{N}|}$ and $T = [A^\top M^\top]^\top$. Therefore, given the initial iterates $\mathbf{x}^0, \boldsymbol{\theta}^0, \boldsymbol{\lambda}^0$ and parameters $\gamma > 0$, $\tau_i, \kappa_i > 0$ for $i \in \mathcal{N}$, choosing $D_x$ and $D_y$ as defined in Definition 1, and setting $\nu_x = \nu_y = 1$, PDA iterations in (5a)-(5b) take the following form:

$$\mathbf{x}^{k+1} \leftarrow \operatorname*{argmin}_{\mathbf{x}} \langle \boldsymbol{\lambda}^k, M\mathbf{x} \rangle + \sum_{i\in\mathcal{N}} \left[ \rho_i(x_i) + \langle \nabla f(x_i^k), x_i \rangle + \langle A_ix_i - b_i, \theta_i^k \rangle + \frac{1}{2\tau_i}\|x_i - x_i^k\|^2 \right], \quad (9a)$$

$$\theta_i^{k+1} \leftarrow \operatorname*{argmin}_{\theta_i} \sigma_{\mathcal{K}_i}(\theta_i) - \langle A_i(2x_i^{k+1} - x_i^k) - b_i, \theta_i \rangle + \frac{1}{2\kappa_i}\|\theta_i - \theta_i^k\|^2, \quad i \in \mathcal{N} \quad (9b)$$

$$\boldsymbol{\lambda}^{k+1} \leftarrow \operatorname*{argmin}_{\boldsymbol{\lambda}} \left\{ -\langle M(2\mathbf{x}^{k+1} - \mathbf{x}^k), \boldsymbol{\lambda} \rangle + \frac{1}{2\gamma}\|\boldsymbol{\lambda} - \boldsymbol{\lambda}^k\|^2 \right\} = \boldsymbol{\lambda}^k + \gamma M(2\mathbf{x}^{k+1} - \mathbf{x}^k). \quad (9c)$$

Since $\mathcal{K}_i$ is a cone, $\mathbf{prox}_{\kappa_i \sigma_{\mathcal{K}_i}}(.) = \mathcal{P}_{\mathcal{K}_i^\circ}(.)$; hence, $\theta_i^{k+1}$ can be written in closed form as

$$\theta_i^{k+1} = \mathcal{P}_{\mathcal{K}_i^\circ}\Big(\theta_i^k + \kappa_i\Big(A_i(2x_i^{k+1} - x_i^k) - b_i\Big)\Big), \quad i \in \mathcal{N}.$$

Using recursion in (9c), we can write $\boldsymbol{\lambda}^{k+1}$ as a partial summation of primal iterates $\{x^\ell\}_{\ell=0}^k$, i.e., $\boldsymbol{\lambda}^k = \boldsymbol{\lambda}^0 + \gamma \sum_{\ell=0}^{k-1} M(2\mathbf{x}^{\ell+1} - \mathbf{x}^\ell)$. Let $\boldsymbol{\lambda}^0 \leftarrow \gamma M\mathbf{x}^0$, $\mathbf{s}^0 \leftarrow \mathbf{x}^0$, and $\mathbf{s}^k \triangleq \mathbf{x}^k + \sum_{\ell=1}^k \mathbf{x}^\ell$ for $k \geq 1$; hence, $\boldsymbol{\lambda}^k = \gamma M\mathbf{s}^k$. Using the fact that $M^\top M = \Omega \otimes \mathbf{I}_n$, we obtain

$$\langle M\mathbf{x}, \boldsymbol{\lambda}^k \rangle = \gamma \langle \mathbf{x}, (\Omega \otimes \mathbf{I}_n)\mathbf{s}^k \rangle = \gamma \sum_{i \in \mathcal{N}} \langle x_i, \sum_{j \in \mathcal{N}_i} (s_i^k - s_j^k) \rangle.$$

Thus, PDA iterations given in (9) for the static graph $\mathcal{G}$ can be computed in a *decentralized* way, via the node-specific computations as in Algorithm DPDA-S displayed in Fig. 1 below.

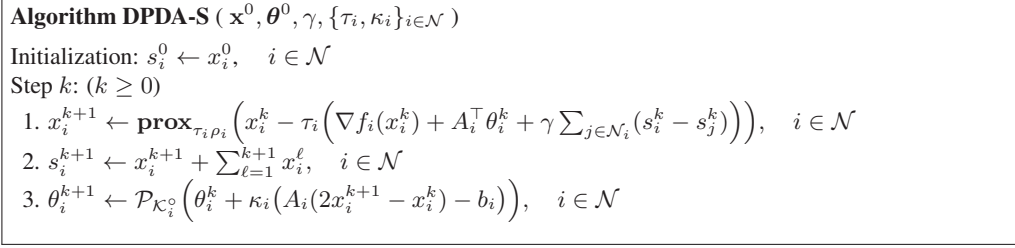

**Figure 1:** Distributed Primal Dual Algorithm for Static $\mathcal{G}$ (DPDA-S)

The convergence rate for DPDA-S, given in (6), follows from Theorem 1.1 with the help of following technical lemma which provides a sufficient condition for $\bar{\mathbf{Q}}(A, A_0) \succ \mathbf{0}$.

**Lemma 2.1.** *Given $\{\tau_i, \kappa_i\}_{i \in \mathcal{N}}$ and $\gamma$ such that $\gamma > 0$, and $\tau_i, \kappa_i > 0$ for $i \in \mathcal{N}$, let $A_0 = M$ and $A \triangleq \mathbf{diag}([A_i]_{i \in \mathcal{N}})$. Then $\bar{\mathbf{Q}} \triangleq \bar{\mathbf{Q}}(A, A_0) \succeq \mathbf{0}$ if $\{\tau_i, \kappa_i\}_{i \in \mathcal{N}}$ and $\gamma$ are chosen such that*

$$\left(\frac{1}{\tau_i} - L_i - 2\gamma d_i\right)\frac{1}{\kappa_i} \geq \sigma_{\max}^2(A_i), \quad \forall i \in \mathcal{N}, \tag{10}$$

*and $\bar{\mathbf{Q}} \succ \mathbf{0}$ if (10) holds with strict inequality, where $\bar{\mathbf{Q}}(A, A_0)$ is defined in Theorem 1.1.*

**Remark 2.1.** *Choosing $\tau_i = (c_i + L_i + 2\gamma d_i)^{-1}$, $\kappa_i = c_i/\sigma_{\max}^2(A_i)$ for any $c_i > 0$ satisfies (10).*

Next, we quantify the suboptimality and infeasibility of the DPDA-S iterate sequence.

**Theorem 2.2.** *Suppose Assumption 1.1 holds. Let $\{\mathbf{x}^k, \boldsymbol{\theta}^k, \boldsymbol{\lambda}^k\}_{k\geq 0}$ be the sequence generated by Algorithm DPDA-S, displayed in Fig. 1, initialized from an arbitrary $\mathbf{x}^0$ and $\boldsymbol{\theta}^0 = \mathbf{0}$. Let step-sizes $\{\tau_i, \kappa_i\}_{i \in \mathcal{N}}$ and $\gamma$ be chosen satisfying (10) with strict inequality. Then $\{\mathbf{x}^k, \boldsymbol{\theta}^k, \boldsymbol{\lambda}^k\}_{k\geq 0}$ converges to $\{\mathbf{x}^*, \boldsymbol{\theta}^*, \boldsymbol{\lambda}^*\}$, a saddle point of (8) such that $\mathbf{x}^* = \mathbf{1} \otimes x^*$ and $(x^*, \boldsymbol{\theta}^*)$ is a primal-dual optimal solution to (3); moreover, the following error bounds hold for all $K \geq 1$:*

$$\|\boldsymbol{\lambda}^*\| \|M\bar{\mathbf{x}}^K\| + \sum_{i \in \mathcal{N}} \|\theta_i^*\| d_{\mathcal{K}_i}(A_i\bar{\mathbf{x}}_i^K - b_i) \leq \Theta_1/K, \qquad |\Phi(\bar{\mathbf{x}}^K) - \Phi(\mathbf{x}^*)| \leq \Theta_1/K,$$

*where $\Theta_1 \triangleq \frac{2}{\gamma}\|\boldsymbol{\lambda}^*\|^2 - \frac{\gamma}{2}\|M\mathbf{x}^0\|^2 + \sum_{i \in \mathcal{N}}\left[\frac{1}{2\tau_i}\|x_i^* - x_i^0\|^2 + \frac{4}{\kappa_i}\|\theta_i^*\|^2\right]$, and $\bar{\mathbf{x}}^K \triangleq \frac{1}{K}\sum_{k=1}^K \mathbf{x}^k$.*

## 3 Dynamic Network Topology

In this section we develop a distributed primal-dual algorithm for solving (3) when the communication network topology is *time-varying*. We assume a *compact* domain, i.e., let $D_i \triangleq \max_{x_i, x_i' \in \mathbf{dom}\,\rho_i} \|x - x'\|$ and $B \triangleq \max_{i \in \mathcal{N}} D_i < \infty$. Let $C$ be the set of consensus decisions:

$$C \triangleq \{\mathbf{x} \in \mathbb{R}^{n|\mathcal{N}|} : x_i = \bar{x}, \forall i \in \mathcal{N} \text{ for some } \bar{x} \in \mathbb{R}^n \text{ s.t. } \|\bar{x}\| \leq B\},$$

then one can reformulate (3) in a decentralized way as follows:

$$\min_{\mathbf{x}} \max_{\mathbf{y}} \mathcal{L}(\mathbf{x}, \mathbf{y}) \triangleq \sum_{i \in \mathcal{N}}\Big(\Phi_i(x_i) + \langle \theta_i, A_i x_i - b_i \rangle - \sigma_{\mathcal{K}_i}(\theta_i)\Big) + \langle \boldsymbol{\lambda}, \mathbf{x} \rangle - \sigma_C(\boldsymbol{\lambda}), \tag{11}$$

where $\mathbf{y} = [\boldsymbol{\theta}^\top \boldsymbol{\lambda}^\top]^\top$ such that $\boldsymbol{\lambda} \in \mathbb{R}^{n|\mathcal{N}|}$, $\boldsymbol{\theta} = [\theta_i]_{i \in \mathcal{N}} \in \mathbb{R}^m$, and $m \triangleq \sum_{i \in \mathcal{N}} m_i$.

Next, we consider the implementation of PDA in (5) to solve (11). Let $\Phi(\mathbf{x}) \triangleq \sum_{i \in \mathcal{N}} \Phi_i(x_i)$, and $h(\mathbf{y}) \triangleq \sigma_{\mathcal{C}}(\boldsymbol{\lambda}) + \sum_{i \in \mathcal{N}} \sigma_{\mathcal{K}_i}(\theta_i) + \langle b_i, \theta_i \rangle$. Define the block-diagonal matrix $A \triangleq \mathbf{diag}([A_i]_{i \in \mathcal{N}}) \in \mathbb{R}^{m \times n |\mathcal{N}|}$ and $T = [A^\top \ \mathbf{I}_{n|\mathcal{N}|}]^\top$. Therefore, given the initial iterates $\mathbf{x}^0, \boldsymbol{\theta}^0, \boldsymbol{\lambda}^0$ and parameters $\gamma > 0, \tau_i, \kappa_i > 0$ for $i \in \mathcal{N}$, choosing $D_x$ and $D_y$ as defined in Definition 1, and setting $\nu_x = \nu_y = 1$, PDA iterations given in (5) take the following form: Starting from $\boldsymbol{\mu}^0 = \boldsymbol{\lambda}^0$, compute for $i \in \mathcal{N}$

$$x_i^{k+1} \leftarrow \operatorname*{argmin}_{\mathbf{x}} \rho_i(x_i) + \langle \nabla f(x_i^k), x_i \rangle + \langle A_i x_i - b_i, \theta_i^k \rangle + \langle x_i, \mu_i^k \rangle + \frac{1}{2\tau_i} \|x_i - x_i^k\|_2^2, \qquad (12a)$$

$$\theta_i^{k+1} \leftarrow \operatorname*{argmin}_{\theta_i} \sigma_{\mathcal{K}_i}(\theta_i) - \langle A_i(2x_i^{k+1} - x_i^k) - b_i, \ \theta_i \rangle + \frac{1}{2\kappa_i} \|\theta_i - \theta_i^k\|_2^2, \qquad (12b)$$

$$\boldsymbol{\lambda}^{k+1} \leftarrow \operatorname*{argmin}_{\boldsymbol{\mu}} \sigma_C(\boldsymbol{\mu}) - \langle 2\mathbf{x}^{k+1} - \mathbf{x}^k, \boldsymbol{\mu} \rangle + \frac{1}{2\gamma} \|\boldsymbol{\mu} - \boldsymbol{\mu}^k\|_2^2, \qquad \boldsymbol{\mu}^{k+1} \leftarrow \boldsymbol{\lambda}^{k+1}. \qquad (12c)$$

Using extended Moreau decomposition for proximal operators, $\boldsymbol{\lambda}^{k+1}$ can be written as

$$\boldsymbol{\lambda}^{k+1} = \operatorname*{argmin}_{\boldsymbol{\mu}} \sigma_C(\boldsymbol{\mu}) + \frac{1}{2\gamma} \|\boldsymbol{\mu} - (\boldsymbol{\mu}^k + \gamma(2\mathbf{x}^{k+1} - \mathbf{x}^k))\|^2 = \mathbf{prox}_{\gamma \sigma_C}(\boldsymbol{\mu}^k + \gamma(2\mathbf{x}^{k+1} - \mathbf{x}^k))$$

$$= \boldsymbol{\mu}^k + \gamma(2\mathbf{x}^{k+1} - \mathbf{x}^k) - \gamma \, \mathcal{P}_C\left(\frac{1}{\gamma}\boldsymbol{\mu}^k + 2\mathbf{x}^{k+1} - \mathbf{x}^k\right). \qquad (13)$$

Let $\mathbf{1} \in \mathbb{R}^{|\mathcal{N}|}$ be the vector all ones, $\mathcal{B}_0 \triangleq \{x \in \mathbb{R}^n : \ \|x\| \le B\}$. Note $\mathcal{P}_{\mathcal{B}_0}(x) = x \min\{1, \frac{B}{\|x\|}\}$. For any $\mathbf{x} = [x_i]_{i \in \mathcal{N}} \in \mathbb{R}^{n|\mathcal{N}|}$, $\mathcal{P}_C(\mathbf{x})$ can be computed as

$$\mathcal{P}_C(\mathbf{x}) = \mathbf{1} \otimes p(\mathbf{x}), \quad \text{where} \quad p(\mathbf{x}) \triangleq \operatorname*{argmin}_{\xi \in \mathcal{B}_0} \sum_{i \in \mathcal{N}} \|\xi - x_i\|^2 = \operatorname*{argmin}_{\xi \in \mathcal{B}_0} \|\xi - \frac{1}{|\mathcal{N}|} \sum_{i \in \mathcal{N}} x_i\|^2. \qquad (14)$$

Let $\mathcal{B} \triangleq \{\mathbf{x} : \ \|x_i\| \le B, \ i \in \mathcal{N}\} = \Pi_{i \in \mathcal{N}} \mathcal{B}_0$. Hence, we can write $\mathcal{P}_C(\mathbf{x}) = \mathcal{P}_\mathcal{B}((W \otimes \mathbf{I}_n)\mathbf{x})$ where $W \triangleq \frac{1}{|\mathcal{N}|} \mathbf{1}\mathbf{1}^\top \in \mathbb{R}^{|\mathcal{N}| \times |\mathcal{N}|}$. Equivalently,

$$\mathcal{P}_C(\mathbf{x}) = \mathcal{P}_\mathcal{B}(\mathbf{1} \otimes \tilde{p}(\mathbf{x})), \quad \text{where} \quad \tilde{p}(\mathbf{x}) \triangleq \frac{1}{|\mathcal{N}|} \sum_{i \in \mathcal{N}} x_i. \qquad (15)$$

Although $\mathbf{x}$-step and $\boldsymbol{\theta}$-step of the PDA implementation in (12) can be computed locally at each node, computing $\boldsymbol{\lambda}^{k+1}$ requires communication among the nodes. Indeed, evaluating the average operator $\tilde{p}(.)$ is *not* a simple operation in a decentralized computational setting which only allows for communication among neighbors. In order to overcome this issue, we will approximate $\tilde{p}(.)$ operator using multi-consensus steps, and analyze the resulting iterations as an *inexact* primal-dual algorithm. In [20], this idea has been exploited within a distributed primal algorithm for unconstrained consensus optimization problems. We define the *consensus step* as one time exchanging local variables among neighboring nodes – the details of this operation will be discussed shortly. Since the connectivity network is dynamic, let $\mathcal{G}^t = (\mathcal{N}, \mathcal{E}^t)$ be the connectivity network at the time $t$-th consensus step is realized for $t \in \mathbb{Z}_+$. We adopt the information exchange model in [21].

**Assumption 3.1.** *Let $V^t \in \mathbb{R}^{|\mathcal{N}| \times |\mathcal{N}|}$ be the weight matrix corresponding to $\mathcal{G}^t = (\mathcal{N}, \mathcal{E}^t)$ at the time of $t$-th consensus step and $\mathcal{N}_i^t \triangleq \{j \in \mathcal{N} : \ (i,j) \in \mathcal{E}^t \text{ or } (j,i) \in \mathcal{E}^t\}$. Suppose for all $t \in \mathbb{Z}_+$: (i) $V^t$ is doubly stochastic; (ii) there exists $\zeta \in (0,1)$ such that for $i \in \mathcal{N}$, $V_{ij}^t \ge \zeta$ if $j \in \mathcal{N}_i^t$, and $V_{ij}^t = 0$ if $j \notin \mathcal{N}_i^t$; (iii) $\mathcal{G}^\infty = (\mathcal{N}, \mathcal{E}^\infty)$ is connected where $\mathcal{E}^\infty \triangleq \{(i,j) \in \mathcal{N} \times \mathcal{N} : (i,j) \in \mathcal{E}^t \text{ for infinitely many } t \in \mathbb{Z}_+\}$, and there exists $\mathbb{Z} \ni T^\circ > 1$ such that if $(i,j) \in \mathcal{E}^\infty$, then $(i,j) \in \mathcal{E}^t \cup \mathcal{E}^{t+1} \cup ... \cup \mathcal{E}^{t+T^\circ-1}$ for all $t \ge 1$.*

**Lemma 3.1.** *[21] Let Assumption 3.1 holds, and $W^{t,s} = V^t V^{t-1} ... V^{s+1}$ for $t \ge s+1$. Given $s \ge 0$ the entries of $W^{t,s}$ converges to $\frac{1}{N}$ as $t \to \infty$ with a geometric rate, i.e., for all $i, j \in \mathcal{N}$, one has $\left| W_{ij}^{t,s} - \frac{1}{N} \right| \le \Gamma \alpha^{t-s}$, where $\Gamma \triangleq 2(1+\zeta^{-\bar{T}})/(1-\zeta^{\bar{T}})$, $\alpha \triangleq (1-\zeta^{\bar{T}})^{1/\bar{T}}$, and $\bar{T} \triangleq (N-1)T^\circ$.*

Consider the $k$-th iteration of PDA as shown in (12). Instead of computing $\boldsymbol{\lambda}^{k+1}$ exactly according to (13), we propose to *approximate* $\boldsymbol{\lambda}^{k+1}$ with the help of Lemma 3.1 and set $\boldsymbol{\mu}^{k+1}$ to this approximation. In particular, let $t_k$ be the total number of consensus steps done before $k$-th iteration of PDA, and let $q_k \ge 1$ be the number of consensus steps within iteration $k$. For $\mathbf{x} = [x_i]_{i \in \mathcal{N}}$, define

$$\mathcal{R}^k(\mathbf{x}) \triangleq \mathcal{P}_\mathcal{B}\left((W^{t_k+q_k,t_k} \otimes \mathbf{I}_n) \mathbf{x}\right) \qquad (16)$$

to approximate $\mathcal{P}_C(\mathbf{x})$ in (13). Note that $\mathcal{R}^k(\cdot)$ can be computed in a *distributed fashion* requiring $q_k$ communications with the neighbors for each node. Indeed,

$$\mathcal{R}^k(\mathbf{x}) = [\mathcal{R}_i^k(\mathbf{x})]_{i \in \mathcal{N}} \quad \text{such that} \quad \mathcal{R}_i^k(\mathbf{x}) \triangleq \mathcal{P}_{\mathcal{B}_0}\left(\sum_{j \in \mathcal{N}} W_{ij}^{t_k+q_k,t_k} x_j\right). \qquad (17)$$

Moreover, the approximation error, $\mathcal{R}^k(\mathbf{x}) - \mathcal{P}_C(\mathbf{x})$, for any $\mathbf{x}$ can be bounded as in (18) due to non-expansivity of $\mathcal{P}_{\mathcal{B}}$ and using Lemma 3.1. From (15), we get for all $i \in \mathcal{N}$,

$$\|\mathcal{R}_i^k(\mathbf{x}) - \mathcal{P}_{\mathcal{B}_0}(\tilde{p}(\mathbf{x}))\| = \|\mathcal{P}_{\mathcal{B}_0}\big(\sum_{j \in \mathcal{N}} W_{ij}^{t_k+q_k,t_k} x_j\big) - \mathcal{P}_{\mathcal{B}_0}\big(\tfrac{1}{N}\sum_{j \in \mathcal{N}} x_j\big)\|$$

$$\leq \|\sum_{j \in \mathcal{N}}\big(W_{ij}^{t_k+q_k,t_k} - \tfrac{1}{N}\big)x_j\| \leq \sqrt{N}\,\Gamma\alpha^{q_k}\|\mathbf{x}\|. \tag{18}$$

Thus, (15) implies that $\|\mathcal{R}^k(\mathbf{x}) - \mathcal{P}_C(\mathbf{x})\| \leq N\,\Gamma\alpha^{q_k}\|\mathbf{x}\|$. Next, to obtain an *inexact* variant of (12), we replace the exact computation in (12c) with the inexact iteration rule:

$$\boldsymbol{\mu}^{k+1} \leftarrow \boldsymbol{\mu}^k + \gamma(2\mathbf{x}^{k+1} - \mathbf{x}^k) - \gamma\mathcal{R}^k\big(\tfrac{1}{\gamma}\boldsymbol{\mu}^k + 2\mathbf{x}^{k+1} - \mathbf{x}^k\big). \tag{19}$$

Thus, PDA iterations given in (12) can be computed inexactly, but in *decentralized* way for dynamic connectivity, via the node-specific computations as in Algorithm DPDA-D displayed in Fig. 2 below.

---

**Algorithm DPDA-D** ( $\mathbf{x}^0, \boldsymbol{\theta}^0, \gamma, \{\tau_i, \kappa_i\}_{i \in \mathcal{N}}, \{q_k\}_{k \geq 0}$ )

Initialization: $\mu_i^0 \leftarrow \mathbf{0}, \quad i \in \mathcal{N}$
Step $k$: ($k \geq 0$)
1. $x_i^{k+1} \leftarrow \mathbf{prox}_{\tau_i \rho_i}\big(x_i^k - \tau_i\big(\nabla f_i(x_i^k) + A_i^\top \theta_i^k + \mu_i^k\big)\big), \quad r_i \leftarrow \tfrac{1}{\gamma}\mu_i^k + 2x_i^{k+1} - x_i^k \quad i \in \mathcal{N}$
2. $\theta_i^{k+1} \leftarrow \mathcal{P}_{\mathcal{K}_i^\circ}\big(\theta_i^k + \kappa_i\big(A_i(2x_i^{k+1} - x_i^k) - b_i\big)\big), \quad i \in \mathcal{N}$
3. For $\ell = 1, \ldots, q_k$
4. $\quad\quad r_i \leftarrow \sum_{j \in \mathcal{N}_i^{t_k+\ell} \cup \{i\}} V_{ij}^{t_k+\ell} r_j, \quad i \in \mathcal{N}$
5. End For
6. $\mu_i^{k+1} \leftarrow \mu_i^k + \gamma(2x_i^{k+1} - x_i^k) - \gamma\mathcal{P}_{\mathcal{B}_0}(r_i), \quad i \in \mathcal{N}$

---

**Figure 2:** Distributed Primal Dual Algorithm for Dynamic $\mathcal{G}^t$ (DPDA-D)

Next, we define the proximal error sequence $\{\mathbf{e}^k\}_{k \geq 1}$ as in (20), which will be used later for analyzing the convergence of Algorithm DPDA-D displayed in Fig. 2.

$$\mathbf{e}^{k+1} \triangleq \mathcal{P}_C\big(\tfrac{1}{\gamma}\boldsymbol{\mu}^k + 2\mathbf{x}^{k+1} - \mathbf{x}^k\big) - \mathcal{R}^k\big(\tfrac{1}{\gamma}\boldsymbol{\mu}^k + 2\mathbf{x}^{k+1} - \mathbf{x}^k\big); \tag{20}$$

hence, $\boldsymbol{\mu}^k = \boldsymbol{\lambda}^k + \gamma\mathbf{e}^k$ for $k \geq 1$ when (12c) is replaced with (19). In the rest, we assume $\boldsymbol{\mu}^0 = \mathbf{0}$. The following observation will also be useful to prove error bounds for DPDA-D iterate sequence. For each $i \in \mathcal{N}$, the definition of $\mathcal{R}_i^k$ in (17) implies that $\mathcal{R}_i^k(\mathbf{x}) \in \mathcal{B}_0$ for all $\mathbf{x}$; hence, from (19),

$$\|\mu_i^{k+1}\| \leq \|\mu_i^k + \gamma(2x_i^{k+1} - x_i^k)\| + \gamma\|\mathcal{R}_i^k\big(\tfrac{1}{\gamma}\boldsymbol{\mu}^k + 2\mathbf{x}^{k+1} - \mathbf{x}^k\big)\| \leq \|\mu_i^k\| + 4\gamma B.$$

Thus, we trivially get the following bound on $\|\boldsymbol{\mu}^k\|$:

$$\|\boldsymbol{\mu}^k\| \leq 4\gamma\sqrt{N}\,B\,k. \tag{21}$$

Moreover, for any $\boldsymbol{\mu}$ and $\boldsymbol{\lambda}$ we have that

$$\sigma_C(\boldsymbol{\mu}) = \sup_{\mathbf{x} \in C} \langle \boldsymbol{\lambda}, \mathbf{x}\rangle + \langle \boldsymbol{\mu} - \boldsymbol{\lambda}, \mathbf{x}\rangle \leq \sigma_C(\boldsymbol{\lambda}) + \sqrt{N}\,B\,\|\boldsymbol{\mu} - \boldsymbol{\lambda}\|. \tag{22}$$

**Theorem 3.2.** *Suppose Assumption 1.1 holds. Starting from $\boldsymbol{\mu}^0 = \mathbf{0}$, $\boldsymbol{\theta}^0 = \mathbf{0}$, and an arbitrary $\mathbf{x}^0$, let $\{\mathbf{x}^k, \boldsymbol{\theta}^k, \boldsymbol{\mu}^k\}_{k \geq 0}$ be the iterate sequence generated using Algorithm DPDA-D, displayed in Fig. 2, using $q_k = \sqrt[p]{k}$ consensus steps at the $k$-th iteration for all $k \geq 1$ for some rational $p \geq 1$. Let primal-dual step-sizes $\{\tau_i, \kappa_i\}_{i \in \mathcal{N}}$ and $\gamma$ be chosen such that the following holds:*

$$\Big(\frac{1}{\tau_i} - L_i - \gamma\Big)\frac{1}{\kappa_i} > \sigma_{\max}^2(A_i), \quad \forall\, i \in \mathcal{N}. \tag{23}$$

*Then $\{\mathbf{x}^k, \boldsymbol{\theta}^k, \boldsymbol{\mu}^k\}_{k \geq 0}$ converges to $\{\mathbf{x}^*, \boldsymbol{\theta}^*, \boldsymbol{\lambda}^*\}$, a saddle point of (11) such that $\mathbf{x}^* = \mathbf{1} \otimes x^*$ and $(x^*, \boldsymbol{\theta}^*)$ is a primal-dual optimal solution to (3). Moreover, the following bounds hold for all $K \geq 1$:*

$$\|\boldsymbol{\lambda}^*\|\,d_{\tilde{C}}(\bar{\mathbf{x}}^K) + \sum_{i \in \mathcal{N}} \|\theta_i^*\|\,d_{\mathcal{K}_i}(A_i\bar{x}_i^K - b_i) \leq \frac{\Theta_2 + \Theta_3(K)}{K}, \qquad |\Phi(\bar{\mathbf{x}}^K) - \Phi(\mathbf{x}^*)| \leq \frac{\Theta_2 + \Theta_3(K)}{K},$$

*where $\bar{\mathbf{x}}^K \triangleq \frac{1}{K}\sum_{k=1}^K \mathbf{x}^k$, $\Theta_2 \triangleq 2\|\boldsymbol{\lambda}^*\|\big(\frac{1}{\gamma}\|\boldsymbol{\lambda}^*\| + \|\mathbf{x}^0 - \mathbf{x}^*\|\big) + \sum_{i \in \mathcal{N}}\big[\frac{1}{\tau_i}\|x_i^* - x_i^0\|^2 + \frac{4}{\kappa_i}\|\theta_i^*\|^2\big]$, and $\Theta_3(K) \triangleq 8N^2B^2\Gamma\sum_{k=1}^K \alpha^{q_k}\big[2\gamma k^2 + \big(\gamma + \frac{\|\boldsymbol{\lambda}^*\|}{\sqrt{N}B}\big)k\big]$. Moreover, $\sup_{K \in \mathbb{Z}_+}\Theta_3(K) < \infty$; hence, $\frac{1}{K}\Theta_3(K) = \mathcal{O}(\frac{1}{K})$.*

**Remark 3.1.** *Note that the suboptimality, infeasibility and consensus violation at the $K$-th iteration is $\mathcal{O}(\Theta_3(K)/K)$, where $\Theta_3(K)$ denotes the error accumulation due to approximation errors, and $\Theta_3(K)$ can be bounded above for all $K \geq 1$ as $\Theta_3(K) \leq R\sum_{k=1}^{K} \alpha^{q_k} k^2$ for some constant $R > 0$. Since $\sum_{k=1}^{\infty} \alpha^{\sqrt[p]{k}} k^2 < \infty$ for any $p \geq 1$, if one chooses $q_k = \sqrt[p]{k}$ for $k \geq 1$, then the total number of communications per node until the end of $K$-th iteration can be bounded above by $\sum_{k=1}^{K} q_k = \mathcal{O}(K^{1+1/p})$. For large $p$, $q_k$ grow slowly, it makes the method more practical at the cost of longer convergence time due to increase in $\mathcal{O}(1)$ constant. Note that $q_k = (\log(k))^2$ also works and it grows very slowly. We assume agents know $q_k$ as a function of $k$ at the beginning, hence, synchronicity can be achieved by simply counting local communications with each neighbor.*

## 4    Numerical Section

We tested DPDA-S and DPDA-D on a primal linear SVM problem where the data is distributed among the computing nodes in $\mathcal{N}$. For the *static* case, communication network $G = (\mathcal{N}, \mathcal{E})$ is a connected graph that is generated by randomly adding edges to a spanning tree, generated uniformly at random, until a desired algebraic connectivity is achieved. For the *dynamic* case, for each consensus round $t \geq 1$, $\mathcal{G}^t$ is generated as in the static case, and $V^t \triangleq \mathbf{I} - \frac{1}{c}\Omega^t$, where $\Omega^t$ is the Laplacian of $\mathcal{G}^t$, and the constant $c > d_{\max}^t$. We ran DPDA-S and DPDA-D on the line and complete graphs as well to see the topology effect – for the dynamic case when the topology is line, each $\mathcal{G}^t$ is a *random line graph*. Let $\mathcal{S} \triangleq \{1, 2, .., s\}$ and $\mathcal{D} \triangleq \{(x_\ell, y_\ell) \in \mathbb{R}^n \times \{-1, +1\} : \ell \in \mathcal{S}\}$ be a set of feature vector and label pairs. Suppose $\mathcal{S}$ is partitioned into $\mathcal{S}_{\text{test}}$ and $\mathcal{S}_{\text{train}}$, i.e., the index sets for the test and training data; let $\{\mathcal{S}_i\}_{i \in \mathcal{N}}$ be a partition of $\mathcal{S}_{\text{train}}$ among the nodes $\mathcal{N}$. Let $\mathbf{w} = [w_i]_{i \in \mathcal{N}}$, $\mathbf{b} = [b_i]_{i \in \mathcal{N}}$, and $\xi \in \mathbb{R}^{|\mathcal{S}_{\text{train}}|}$ such that $w_i \in \mathbb{R}^n$ and $b_i \in \mathbb{R}$ for $i \in \mathcal{N}$.

Consider the following distributed SVM problem:

$$\min_{\mathbf{w}, \mathbf{b}, \xi} \left\{ \frac{1}{2} \sum_{i \in \mathcal{N}} \|w_i\|^2 + C |\mathcal{N}| \sum_{i \in \mathcal{N}} \sum_{\ell \in \mathcal{S}_i} \xi_\ell : \begin{array}{cc} y_\ell(w_i^T x_\ell + b_i) \geq 1 - \xi_\ell, & \xi_\ell \geq 0, \quad \ell \in \mathcal{S}_i, \ i \in \mathcal{N}, \\ w_i = w_j, & b_i = b_j \quad (i,j) \in \mathcal{E} \end{array} \right\}$$

Similar to [3], $\{x_\ell\}_{\ell \in \mathcal{S}}$ is generated from two-dimensional multivariate Gaussian distribution with covariance matrix $\Sigma = [1, 0; 0, 2]$ and with mean vector either $m_1 = [-1, -1]^T$ or $m_2 = [1, 1]^T$ with equal probability. The experiment was performed for $C = 2$, $|\mathcal{N}| = 10$, $s = 900$ such that $|\mathcal{S}_{\text{test}}| = 600$, $|\mathcal{S}_i| = 30$ for $i \in \mathcal{N}$, i.e., $|\mathcal{S}_{\text{train}}| = 300$, and $q_k = \sqrt{k}$. We ran DPDA-S and DPDA-D on line, random, and complete graphs, where the random graph is generated such that the algebraic connectivity is approximately 4. Relative suboptimality and relative consensus violation, i.e., $\max_{(i,j) \in \mathcal{E}} \|[w_i^\top b_i]^\top - [w_j^\top b_j]^\top\| / \left\|[w^{*\top} b^*]\right\|$, and absolute feasibility violation are plotted against iteration counter in Fig. 3, where $[w^{*\top} b^*]$ denotes the optimal solution to the central problem. As expected, the convergence is slower when the connectivity of the graph is weaker. Furthermore, visual comparison between DPDA-S, local SVMs (for two nodes) and centralized SVM for the same training and test data sets is given in Fig. 4 and Fig. 5 in the appendix.

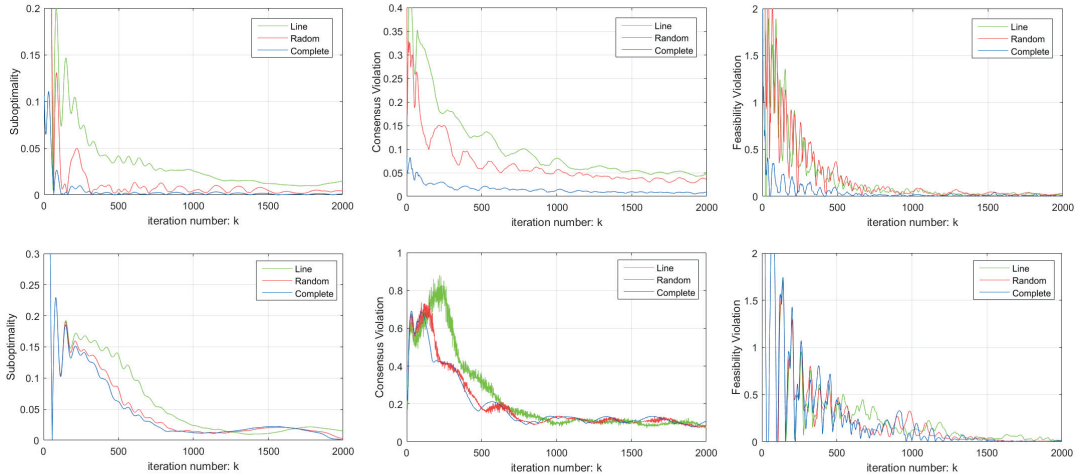

**Figure 3:** Static (top) and Dynamic (bottom) network topologies: line, random, and complete graphs

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
