[Supplementary Material · NIPS_final_supplementary_v2.pdf]

# 5 Appendix

## 5.1 Proof of Theorem 1.1

**Lemma 5.1.** *Let* $\bar{\mathbf{Q}} \triangleq \bar{\mathbf{Q}}(A, A_0)$*, and* $\mathbf{z} = [\mathbf{x}^\top \mathbf{y}^\top]^\top$ *for* $\mathbf{x} \in \mathcal{X}$, $\mathbf{y} \in \mathcal{Y}$*. For any* $\mathbf{x} \in \mathcal{X}$*, and* $\mathbf{y} \in \mathcal{Y}$*, the iterate sequence* $\{\mathbf{z}^k\}_{k \geq 1}$ *defined as in Theorem 1.1 satisfies for all* $k \geq 0$

$$
\mathcal{L}(\mathbf{x}^{k+1}, \mathbf{y}) - \mathcal{L}(\mathbf{x}, \mathbf{y}^{k+1}) \leq \left[ D_x(\mathbf{x}, \mathbf{x}^k) + D_y(\mathbf{y}, \mathbf{y}^k) - \left\langle T(\mathbf{x} - \mathbf{x}^k), \ \mathbf{y} - \mathbf{y}^k \right\rangle \right]
$$
$$
- \left[ D_x(\mathbf{x}, \mathbf{x}^{k+1}) + D_y(\mathbf{y}, \mathbf{y}^{k+1}) - \left\langle T(\mathbf{x} - \mathbf{x}^{k+1}), \ \mathbf{y} - \mathbf{y}^{k+1} \right\rangle \right] - \frac{1}{2}(\mathbf{z}^{k+1} - \mathbf{z}^k)^\top \bar{\mathbf{Q}}(\mathbf{z}^{k+1} - \mathbf{z}^k).
\tag{24}
$$

*Proof.* Note that x-subproblem in (5a) is separable in local decisions $\{x_i\}_{i \in \mathcal{N}}$; for each $i \in \mathcal{N}$ the local subproblem over $x_i$ is strongly convex with constant $1/\tau_i$. Indeed, let $\mathbf{p}^k = T^\top \mathbf{y}^k$ and define $\{p_i^k\}_{i \in \mathcal{N}}$ such that $p_i^k$ is the subvector corresponding to the components of $x_i$, i.e., $\mathbf{p}^k = [p_i^k]_{i \in \mathcal{N}}$. Thus, the definitions of $\rho$, $f$ and $D_x$, $\nu_x = 1$, and (5a) imply that for all $i \in \mathcal{N}$

$$
x_i^{k+1} = \operatorname*{argmin}_{x_i} \rho_i(x_i) + f_i(x_i^k) + \left\langle \nabla f_i(x_i^k), \ x_i - x_i^k \right\rangle + \left\langle p_i^k, x_i \right\rangle + \frac{1}{2\tau_i} \left\| x_i - x_i^k \right\|^2.
\tag{25}
$$

Therefore, the strong convexity of the objective in local subproblem (25) for $i \in \mathcal{N}$ implies

$$
\rho_i(x_i) + \left\langle \nabla f_i(x_i^k), \ x_i \right\rangle + \left\langle p_i^k, x_i \right\rangle + \frac{1}{2\tau_i} \left\| x_i - x_i^k \right\|^2 \geq
$$
$$
\rho_i(x_i^{k+1}) + \left\langle \nabla f_i(x_i^k), \ x_i^{k+1} \right\rangle + \left\langle p_i^k, x_i^{k+1} \right\rangle + \frac{1}{2\tau_i} \left\| x_i^{k+1} - x_i^k \right\|^2 + \frac{1}{2\tau_i} \left\| x_i - x_i^{k+1} \right\|^2.
$$

Convexity of $f_i$ and Lipschitz continuity of $\nabla f_i$ implies that

$$
f_i(x_i) \geq f_i(x_i^k) + \left\langle \nabla f_i(x_i^k), \ x_i - x_i^k \right\rangle \geq f_i(x_i^{k+1}) + \left\langle \nabla f_i(x_i^k), \ x_i - x_i^{k+1} \right\rangle - \frac{L_i}{2} \left\| x_i^{k+1} - x_i^k \right\|^2.
$$

Since $\sum_{i \in \mathcal{N}} \left\langle p_i^k, \ x_i \right\rangle = \left\langle T\mathbf{x}, \ \mathbf{y}^k \right\rangle$ for all $\mathbf{x}$, summing these two inequalities for each $i \in \mathcal{N}$, and then summing the resulting inequalities over $i \in \mathcal{N}$, we get

$$
\Phi(\mathbf{x}) + D_x(\mathbf{x}, \mathbf{x}^k) \geq
\tag{26}
$$
$$
\Phi(\mathbf{x}^{k+1}) + \left\langle T(\mathbf{x}^{k+1} - \mathbf{x}), \ \mathbf{y}^k \right\rangle + D_x(\mathbf{x}, \mathbf{x}^{k+1}) + \tfrac{1}{2}(\mathbf{x}^{k+1} - \mathbf{x})^\top \bar{\mathbf{D}}_\tau (\mathbf{x}^{k+1} - \mathbf{x}^k).
$$

Similarly, let $\mathbf{q}^k = T(2\mathbf{x}^{k+1} - \mathbf{x}^k)$ and define $q_0^k \in \mathbb{R}^{m_0}$ and $q_i^k \in \mathbb{R}^{m_i}$ for $i \in \mathcal{N}$ such that $q_0^k$ is the subvector corresponding to the components of $\boldsymbol{\lambda}$, and $q_i^k$ is the subvector corresponding to the components of $\theta_i$ for $i \in \mathcal{N}$, i.e., $\mathbf{p}^k = [p_1^{k^\top} \ldots p_N^{k^\top} p_0^{k^\top}]^\top$. Thus, the definitions of $h$ and $D_y$, and $\nu_y = 1$ imply that according to (5b) we have

$$
\boldsymbol{\lambda}^{k+1} = \operatorname*{argmin}_{\boldsymbol{\lambda}} h_0(\boldsymbol{\lambda}) - \left\langle q_0^k, \boldsymbol{\lambda} \right\rangle + \frac{1}{2\gamma} \left\| \boldsymbol{\lambda} - \boldsymbol{\lambda}^k \right\|^2,
$$
$$
\theta_i^{k+1} = \operatorname*{argmin}_{\theta_i} h_i(\theta_i) - \left\langle q_i^k, \theta_i \right\rangle + \frac{1}{2\kappa_i} \left\| \theta_i - \theta_i^k \right\|^2, \quad \forall \, i \in \mathcal{N}.
$$

Therefore, the strong convexity of the objectives in these subproblems implies that

$$
h_0(\boldsymbol{\lambda}) - \left\langle q_0^k, \boldsymbol{\lambda} \right\rangle + \frac{1}{2\gamma} \left\| \boldsymbol{\lambda} - \boldsymbol{\lambda}^k \right\|^2 \geq h_0(\boldsymbol{\lambda}^{k+1}) - \left\langle q_0^k, \boldsymbol{\lambda}^{k+1} \right\rangle + \frac{1}{2\gamma} \left\| \boldsymbol{\lambda}^{k+1} - \boldsymbol{\lambda}^k \right\|^2 + \frac{1}{2\gamma} \left\| \boldsymbol{\lambda} - \boldsymbol{\lambda}^{k+1} \right\|^2,
$$
$$
h_i(\theta_i) - \left\langle q_i^k, \theta_i \right\rangle + \frac{1}{2\kappa_i} \left\| \theta_i - \theta_i^k \right\|^2 \geq h_i(\theta_i^{k+1}) - \left\langle q_i^k, \theta_i^{k+1} \right\rangle + \frac{1}{2\kappa_i} \left\| \theta_i^{k+1} - \theta_i^k \right\|^2 + \frac{1}{2\kappa_i} \left\| \theta_i - \theta_i^{k+1} \right\|^2.
$$

Since $\left\langle q_0^k, \boldsymbol{\lambda} \right\rangle + \sum_{i \in \mathcal{N}} \left\langle q_i^k, \theta_i \right\rangle = \left\langle T(2\mathbf{x}^{k+1} - \mathbf{x}^k), \ \mathbf{y} \right\rangle$ for all $\mathbf{y}$, summing the second inequality over $i \in \mathcal{N}$ and then summing the resulting inequality with the first one, we get

$$
h(\mathbf{y}) + D_y(\mathbf{y}, \mathbf{y}^k) \geq
\tag{27}
$$
$$
h(\mathbf{y}^{k+1}) - \left\langle T(2\mathbf{x}^{k+1} - \mathbf{x}^k), \ \mathbf{y}^{k+1} - \mathbf{y} \right\rangle + D_y(\mathbf{y}, \mathbf{y}^{k+1}) + D_y(\mathbf{y}^{k+1}, \mathbf{y}^k).
$$

Summing (26) and (27) gives the desired result. □

Now we continue to the proof of Theorem 1.1. Let $\bar{\mathbf{Q}} \triangleq \bar{\mathbf{Q}}(A, A_0)$. Since $\bar{\mathbf{Q}} \succeq 0$, we can drop the last term in the inequality given in the statement of Lemma 5.1; and summing it over $k$, we get

$$\sum_{k=0}^{K-1} \mathcal{L}(\mathbf{x}^{k+1}, \mathbf{y}) - \mathcal{L}(\mathbf{x}, \mathbf{y}^{k+1}) \leq \left[ D_x(\mathbf{x}, \mathbf{x}^0) + D_y(\mathbf{y}, \mathbf{y}^0) - \langle T(\mathbf{x} - \mathbf{x}^0), \, \mathbf{y} - \mathbf{y}^0 \rangle \right]$$
$$- \left[ D_x(\mathbf{x}, \mathbf{x}^K) + D_y(\mathbf{y}, \mathbf{y}^K) - \langle T(\mathbf{x} - \mathbf{x}^K), \, \mathbf{y} - \mathbf{y}^K \rangle \right].$$

$\bar{\mathbf{Q}} \succeq 0$ also implies that $D_x(\mathbf{x}, \mathbf{x}^K) + D_y(\mathbf{y}, \mathbf{y}^K) - \langle T(\mathbf{x} - \mathbf{x}^K), \, \mathbf{y} - \mathbf{y}^K \rangle \geq 0$; therefore, (6) follows from Jensen's inequality.

Now suppose $\bar{\mathbf{Q}} \succ 0$, and let $\mathbf{z}^* = [\mathbf{x}^{*\top} \mathbf{y}^{*\top}]^\top$ be a saddle point for (4). From the definition of $\bar{\mathbf{Q}}$, for all $\mathbf{z}, \mathbf{z}'$, we have

$$D_x(\mathbf{x}, \mathbf{x}') + D_y(\mathbf{y}, \mathbf{y}') - \langle T(\mathbf{x} - \mathbf{x}'), \, \mathbf{y} - \mathbf{y}' \rangle \geq \tfrac{1}{2} \|\mathbf{z} - \mathbf{z}'\|_{\bar{\mathbf{Q}}}^2. \tag{28}$$

Evaluating (24) at $\mathbf{z} = \mathbf{z}^*$, we get $k \geq 0$

$$0 \leq \mathcal{L}(\mathbf{x}^{k+1}, \mathbf{y}^*) - \mathcal{L}(\mathbf{x}^*, \mathbf{y}^{k+1}) \leq \left[ D_x(\mathbf{x}^*, \mathbf{x}^k) + D_y(\mathbf{y}^*, \mathbf{y}^k) - \langle T(\mathbf{x}^* - \mathbf{x}^k), \, \mathbf{y}^* - \mathbf{y}^k \rangle \right]$$
$$- \left[ D_x(\mathbf{x}^*, \mathbf{x}^{k+1}) + D_y(\mathbf{y}^*, \mathbf{y}^{k+1}) - \langle T(\mathbf{x}^* - \mathbf{x}^{k+1}), \, \mathbf{y}^* - \mathbf{y}^{k+1} \rangle \right]$$
$$- \tfrac{1}{2} \left\| \mathbf{z}^{k+1} - \mathbf{z}^k \right\|_{\bar{\mathbf{Q}}}^2. \tag{29}$$

Note that (29) implies that $\left\{ D_x(\mathbf{x}^*, \mathbf{x}^k) + D_y(\mathbf{y}^*, \mathbf{y}^k) - \langle T(\mathbf{x}^* - \mathbf{x}^k), \, \mathbf{y}^* - \mathbf{y}^k \rangle \right\}_{k \geq 0}$ is a non-increasing sequence. Using this fact together with (28), we get for all $k \geq 0$

$$\tfrac{1}{2} \left\| \mathbf{z}^{k+1} - \mathbf{z}^* \right\|_{\bar{\mathbf{Q}}}^2 \leq D_x(\mathbf{x}^*, \mathbf{x}^0) + D_y(\mathbf{y}^*, \mathbf{y}^0) - \langle T(\mathbf{x}^* - \mathbf{x}^0), \, \mathbf{y}^* - \mathbf{y}^0 \rangle.$$

Therefore, both $\{\mathbf{z}^k\}$ and $\{\bar{\mathbf{z}}^k\}$ are bounded sequences. Hence, there is a subsequence $\{\mathbf{z}^{k_n}\}_{n \geq 1}$ converging to a limit point $\hat{\mathbf{z}}$. From (29), it follows that $\sum_{k=0}^{\infty} \left\| \mathbf{z}^{k+1} - \mathbf{z}^k \right\|_{\bar{\mathbf{Q}}}^2 < \infty$. Since $\bar{\mathbf{Q}} \succ \mathbf{0}$, for any $\epsilon > 0$, there exists $N_1$ such that for all $n \geq N_1$, we have $\left\| \mathbf{z}^{k_n+1} - \mathbf{z}^{k_n} \right\| < \tfrac{\epsilon}{2}$. From the fact that $\mathbf{z}^{k_n} \to \hat{\mathbf{z}}$, there exists $N_2$ such that for all $n \geq N_2$, we have $\left\| \mathbf{z}^{k_n} - \hat{\mathbf{z}} \right\| < \tfrac{\epsilon}{2}$. Therefore, by letting $N = \max\{N_1, N_2\}$ we get $\left\| \mathbf{z}^{k_n+1} - \hat{\mathbf{z}} \right\| < \epsilon$, i.e., $\mathbf{z}^{k_n+1} \to \hat{\mathbf{z}}$.

The optimality conditions for (5a) and (5b) imply that for all $n \in \mathbb{Z}_+$, we have $\boldsymbol{\alpha}^n \in \partial\rho(\mathbf{x}^{k_n+1})$ and $\boldsymbol{\beta}^n \in \partial h(\mathbf{y}^{k_n+1})$, where

$$\boldsymbol{\alpha}^n \triangleq \nabla\psi_x(\mathbf{x}^{k_n}) - \nabla\psi_x(\mathbf{x}^{k_n+1}) - \left( \nabla f(\mathbf{x}^{k_n}) + T^\top \mathbf{y}^{k_n} \right),$$
$$\boldsymbol{\beta}^n \triangleq \nabla\psi_y(\mathbf{y}^{k_n}) - \nabla\psi_y(\mathbf{y}^{k_n+1}) + T\left( 2\mathbf{x}^{k_n+1} - \mathbf{x}^{k_n} \right).$$

Since $\nabla\psi_x$ and $\nabla\psi_y$ are continuously differentiable on $\mathbf{dom}\,\rho$ and $\mathbf{dom}\,h$, respectively, and since $\rho$ and $h$ are proper, closed convex functions, it follows from Theorem 24.4 in [22] that

$$\partial\rho(\hat{\mathbf{x}}) \ni \lim_n \boldsymbol{\alpha}^n = -\nabla f(\hat{\mathbf{x}}) - T^\top \hat{\mathbf{y}}, \text{ and } \partial h(\hat{\mathbf{y}}) \ni \lim_n \boldsymbol{\beta}^n = T\hat{\mathbf{x}},$$

which also implies that $\hat{\mathbf{z}}$ is a saddle point of (4).

Since (29) is true for any saddle point $\mathbf{z}^*$, by setting $\mathbf{z}^* = \hat{\mathbf{z}}$ in (29), one can conclude that $\{\mathbf{s}^k\}_{k \geq 0}$ is a *nonincreasing* sequence, where

$$\mathbf{s}^k \triangleq D_x(\hat{\mathbf{x}}, \mathbf{x}^k) + D_y(\hat{\mathbf{y}}, \mathbf{y}^k) - \langle T(\hat{\mathbf{x}} - \mathbf{x}^k), \hat{\mathbf{y}} - \mathbf{y}^k \rangle; \tag{30}$$

moreover, $\mathbf{s} \triangleq \lim_k \mathbf{s}^k \geq 0$ exists since $\mathbf{s}^k \geq 0$ for all $k \geq 0$ due to (28). Therefore, $\mathbf{s} = \lim_n \mathbf{s}^{k_n}$; and since $\lim_n \langle T(\hat{\mathbf{x}} - \mathbf{x}^{k_n}), \hat{\mathbf{y}} - \mathbf{y}^{k_n} \rangle = 0$ (from $\mathbf{z}^{k_n} \to \hat{\mathbf{z}}$),

$$\mathbf{s} = \lim_{n \to \infty} D_x(\hat{\mathbf{x}}, \mathbf{x}^{k_n}) + D_y(\mathbf{y}^*, \mathbf{y}^{k_n}) = 0.$$

Therefore, $\mathbf{z}^k \to \hat{\mathbf{z}}$ follows from (28) and (30).

## 5.2 Proof of Lemma 2.1

Since $\mathbf{D}_\gamma \succ 0$, Schur complement condition implies that $\bar{\mathbf{Q}} \succeq 0$ if and only if

$$B - \gamma \begin{bmatrix} M^\top M & 0 \\ 0 & 0 \end{bmatrix} \succeq 0, \quad \text{where} \quad B \triangleq \begin{bmatrix} \bar{\mathbf{D}}_\tau & -A^\top \\ -A & \mathbf{D}_\kappa \end{bmatrix}. \tag{31}$$

Moreover, since $\mathbf{D}_\kappa \succ 0$, again using Schur complement and the fact that $M^\top M = \Omega \otimes \mathbf{I}_n$, one can conclude that (31) holds if and only if $\bar{\mathbf{D}}_\tau - \gamma(\Omega \otimes \mathbf{I}_n) - A^\top \mathbf{D}_\kappa^{-1} A \succeq 0$. By definition $\Omega = \mathbf{diag}([d_i]_{i\in\mathcal{N}}) - E$, where $E_{ii} = 0$ for all $i \in \mathcal{N}$ and $E_{ij} = E_{ji} = 1$ if $(i,j) \in \mathcal{E}$ or $(j,i) \in \mathcal{E}$. Note that $\mathbf{diag}([d_i]_{i\in\mathcal{N}}) + E \succeq 0$ since it is diagonally dominant. Therefore, $\Omega \preceq 2\,\mathbf{diag}([d_i]_{i\in\mathcal{N}})$. Hence, it is sufficient to have $\bar{\mathbf{D}}_\tau - 2\gamma\,\mathbf{diag}([d_i]_{i\in\mathcal{N}}) \otimes \mathbf{I}_n - A^\top \mathbf{D}_\kappa^{-1} A \succeq 0$, and this condition holds if (10) is true. By the same argument, if (10) holds with strict inequality, then $\bar{\mathbf{Q}} \succ \mathbf{0}$.

## 5.3 Proof of Theorem 2.2

We start the proof with a simple observation. Every closed convex cone $\mathcal{C} \in \mathbb{R}^m$ induces a decomposition on $\mathbb{R}^m$, i.e., according to Moreau decomposition, for any $y \in \mathbb{R}^m$, there exist $y^1, y^2 \in \mathbb{R}^m$ such that $y^1 \perp y^2$ and $y = y^1 + y^2$; in particular, $y^1 = \mathcal{P}_\mathcal{C}(y)$ and $y^2 = \mathcal{P}_{\mathcal{C}^\circ}(y)$ where $\mathcal{C}^\circ = -\mathcal{C}^*$ is the polar cone of $\mathcal{C}$. Hence, from the definition of a support function and the fact that $\langle y, w \rangle \leq 0$ for any $y \in \mathcal{C}^\circ$ and $w \in \mathcal{C}$, one can conclude that

$$\sigma_\mathcal{C}(y) = \begin{cases} 0 & y \in \mathcal{C}^\circ \\ +\infty & \text{o.w.} \end{cases} \tag{32}$$

Note the iterate sequence $\{\mathbf{x}^k, \boldsymbol{\theta}^k\}_{k\geq 0}$ generated by Algorithm DPDA-S in Fig. 1 is the same as the PDA iterate sequence $\{\mathbf{x}^k, \boldsymbol{\theta}^k, \boldsymbol{\lambda}^{\bar{k}}\}_{k\geq 0}$ computed according to (9) for solving (8) when $\boldsymbol{\lambda}^0 = \gamma M\mathbf{x}^0$. From Lemma 2.1, since the step-size parameters $\{\tau_i, \kappa_i\}_{i\in\mathcal{N}}$ and $\gamma$ are chosen satisfying (10) with strict inequality, the condition $\bar{\mathbf{Q}}(A, A_0) \succ 0$ in Theorem 1.1 holds, where $A_0 = M$ for problem (8). Therefore, Theorem 1.1 implies that (6) holds for all $K \geq 1$ with $\nu_x = \nu_y = 1$ and Bregman function $D_x, D_y$ defined as in Definition 1. In particular, the result of Theorem 1.1 can be written more explicitly for (8) as follows: let $\bar{\mathbf{x}}^K \triangleq \frac{1}{K}\sum_{k=1}^K \mathbf{x}_k$, $\bar{\boldsymbol{\theta}}^K \triangleq \frac{1}{K}\sum_{k=1}^K \boldsymbol{\theta}^k$ and $\bar{\boldsymbol{\lambda}}^K \triangleq \frac{1}{K}\sum_{k=1}^K \boldsymbol{\lambda}^k$, then for any $\mathbf{x} \in \mathbb{R}^{n|\mathcal{N}|}$, $\boldsymbol{\lambda} \in \mathbb{R}^{n|\mathcal{E}|}$, $\boldsymbol{\theta} \in \mathbb{R}^m$ for $m = \sum_{i\in\mathcal{N}} m_i$, and for all $K \geq 1$, we have

$$\mathcal{L}(\bar{\mathbf{x}}^K, \boldsymbol{\theta}, \boldsymbol{\lambda}) - \mathcal{L}(\mathbf{x}, \bar{\boldsymbol{\theta}}^K, \bar{\boldsymbol{\lambda}}^K) \leq \Theta(\mathbf{x}, \boldsymbol{\theta}, \boldsymbol{\lambda})/K, \tag{33}$$

$$\Theta(\mathbf{x}, \boldsymbol{\theta}, \boldsymbol{\lambda}) \triangleq \frac{1}{2\gamma}\|\boldsymbol{\lambda} - \boldsymbol{\lambda}^0\|^2 - \langle M(\mathbf{x} - \mathbf{x}^0), \boldsymbol{\lambda} - \boldsymbol{\lambda}^0 \rangle$$
$$+ \sum_{i\in\mathcal{N}} \left[ \frac{1}{2\tau_i}\|x_i - x_i^0\|^2 + \frac{1}{2\kappa_i}\|\theta_i - \theta_i^0\|^2 - \langle A_i(x_i - x_i^0), \theta_i - \theta_i^0 \rangle \right].$$

Note that under the assumption in (10), Schur complement condition guarantees that

$$\begin{bmatrix} \frac{1}{\tau_i}\mathbf{I}_n & -A_i^\top \\ -A_i & \frac{1}{\kappa_i}\mathbf{I}_{m_i} \end{bmatrix} \preceq \begin{bmatrix} \frac{2}{\tau_i}\mathbf{I}_n & \mathbf{0}^\top \\ \mathbf{0} & \frac{2}{\kappa_i}\mathbf{I}_{m_i} \end{bmatrix}.$$

Therefore,

$$\Theta(\mathbf{x}, \boldsymbol{\theta}, \boldsymbol{\lambda}) \leq \sum_{i\in\mathcal{N}} \left[ \frac{1}{\tau_i}\|x_i - x_i^0\|^2 + \frac{1}{\kappa_i}\|\theta_i - \theta_i^0\|^2 \right] + \frac{1}{2\gamma}\|\boldsymbol{\lambda} - \boldsymbol{\lambda}^0\|^2 - \langle M(\mathbf{x} - \mathbf{x}^0), \boldsymbol{\lambda} - \boldsymbol{\lambda}^0 \rangle. \tag{34}$$

Note that, if Assumption 1.1 holds, one can construct a primal-dual optimal solution $(\mathbf{x}^*, \boldsymbol{\theta}^*, \boldsymbol{\lambda}^*)$ to (7) which is a saddle point for $\mathcal{L}$ in (8); hence, $\mathcal{L}(\mathbf{x}^*, \boldsymbol{\theta}^*, \boldsymbol{\lambda}^*) = \Phi(\mathbf{x}^*)$ and $\theta_i^* \in \mathcal{K}_i^\circ$ for $i \in \mathcal{N}$. Define $\tilde{\mathbf{w}} = [\tilde{w}_i]_{i\in\mathcal{N}}$ such that $\tilde{w}_i \triangleq A_i\bar{x}_i^K - b_i \in \mathbb{R}^{m_i}$ for $i \in \mathcal{N}$. Since $\mathcal{K}_i$ is a closed convex cone, it induces a decomposition on $\mathbb{R}^{m_i}$ for $i \in \mathcal{N}$, i.e., consider $\tilde{w}_i^1 = \mathcal{P}_{\mathcal{K}_i}(\tilde{w}_i)$ and $\tilde{w}_i^2 = \mathcal{P}_{\mathcal{K}_i^\circ}(\tilde{w}_i)$. Note that since $\tilde{w}_i = \tilde{w}_i^1 + \tilde{w}_i^2$, $\|\tilde{w}_i^2\| = \|\mathcal{P}_{\mathcal{K}_i}(\tilde{w}_i) - \tilde{w}_i\| = d_{\mathcal{K}_i}(\tilde{w}_i)$. Define $\tilde{\boldsymbol{\theta}} = [\tilde{\theta}_i]_{i\in\mathcal{N}}$ such that $\tilde{\theta}_i \triangleq 2\|\theta_i^*\|\frac{1}{\|\tilde{w}_i^2\|}\tilde{w}_i^2 \in \mathcal{K}_i^\circ$. Therefore,

$$\langle A_i\bar{x}_i^K - b_i, \tilde{\theta}_i \rangle = 2\frac{\|\theta_i^*\|}{\|\tilde{w}_i^2\|}\langle \tilde{w}_i^1 + \tilde{w}_i^2, \tilde{w}_i^2 \rangle = 2\|\theta_i^*\|\,d_{\mathcal{K}_i}(A_i\bar{x}_i^K - b_i), \tag{35}$$

where the second equality follows from $\tilde{w}_i^1 \perp \tilde{w}_i^2$. Similarly, define $\tilde{\boldsymbol{\lambda}} \triangleq 2\|\boldsymbol{\lambda}^*\|(M\bar{\mathbf{x}}^K / \|M\bar{\mathbf{x}}^K\|)$. Hence, $\langle M\bar{\mathbf{x}}^K, \tilde{\boldsymbol{\lambda}} \rangle = 2\|\boldsymbol{\lambda}^*\|\|M\bar{\mathbf{x}}^K\|$. Therefore, together with (35), we get

$$\mathcal{L}(\bar{\mathbf{x}}^K, \tilde{\boldsymbol{\theta}}, \tilde{\boldsymbol{\lambda}}) - \mathcal{L}(\mathbf{x}^*, \boldsymbol{\theta}^*, \boldsymbol{\lambda}^*) = \Phi(\bar{\mathbf{x}}^K) - \Phi(\mathbf{x}^*) + 2\left(\|\boldsymbol{\lambda}^*\|\|M\bar{\mathbf{x}}^K\| + \sum_{i \in \mathcal{N}} d_{\mathcal{K}_i}(A_i\bar{x}_i^K - b_i)\|\theta_i^*\|\right).$$

Now we are going to upper bound $\Theta(\mathbf{x}^*, \tilde{\boldsymbol{\theta}}, \tilde{\boldsymbol{\lambda}})$ using (34). Since $\boldsymbol{\lambda}^0 = \gamma M\mathbf{x}^0$, we get

$$\frac{1}{2\gamma}\|\tilde{\boldsymbol{\lambda}} - \boldsymbol{\lambda}^0\|^2 - \langle M(\mathbf{x}^* - \mathbf{x}^0), \tilde{\boldsymbol{\lambda}} - \boldsymbol{\lambda}^0 \rangle = \frac{1}{2\gamma}\left(\|\tilde{\boldsymbol{\lambda}} - \boldsymbol{\lambda}^0 - \gamma M(\mathbf{x}^* - \mathbf{x}^0)\|^2 - \gamma^2\|M(\mathbf{x}^* - \mathbf{x}^0)\|^2\right)$$
$$= \frac{2}{\gamma}\|\boldsymbol{\lambda}^*\|^2 - \frac{\gamma}{2}\|M\mathbf{x}^0\|^2, \tag{36}$$

where in the last equality follows from $M\mathbf{x}^* = 0$. Since $\boldsymbol{\theta}^*$ and $\boldsymbol{\lambda}^*$ maximize the Lagrangian function at $\mathbf{x}^*$, and we set $\theta_i^0 = \mathbf{0}$, the definitions of $\tilde{\theta}_i$, $\tilde{\boldsymbol{\lambda}}$, and (33), (34) together imply that

$$\mathcal{L}(\bar{\mathbf{x}}^K, \tilde{\boldsymbol{\theta}}, \tilde{\boldsymbol{\lambda}}) - \mathcal{L}(\mathbf{x}^*, \boldsymbol{\theta}^*, \boldsymbol{\lambda}^*) \leq \mathcal{L}(\bar{\mathbf{x}}^K, \tilde{\boldsymbol{\theta}}, \tilde{\boldsymbol{\lambda}}) - \mathcal{L}(\mathbf{x}^*, \bar{\boldsymbol{\theta}}^K, \bar{\boldsymbol{\lambda}}^K) \leq \frac{1}{K}\Theta(\mathbf{x}^*, \tilde{\boldsymbol{\theta}}, \tilde{\boldsymbol{\lambda}}) \leq \frac{\Theta_1}{K}.$$

Therefore, we can conclude that

$$\Phi(\bar{\mathbf{x}}^K) - \Phi(\mathbf{x}^*) + 2\left(\|\boldsymbol{\lambda}^*\|\|M\bar{\mathbf{x}}^K\| + \sum_{i \in \mathcal{N}} d_{\mathcal{K}_i}(A_i\bar{x}_i^K - b_i)\|\theta_i^*\|\right) \leq \frac{\Theta_1}{K}, \tag{37}$$

where we use $\mathcal{L}(\mathbf{x}^*, \boldsymbol{\theta}^*, \boldsymbol{\lambda}^*) = \Phi(\mathbf{x}^*)$ and the fact that $\sigma_{\mathcal{K}_i}(\tilde{\theta}_i) = 0$ due to (32) since $\tilde{\theta}_i \in \mathcal{K}_i^\circ$ for $i \in \mathcal{N}$. Moreover, since $(\mathbf{x}^*, \boldsymbol{\theta}^*, \boldsymbol{\lambda}^*)$ is a saddle-point for $\mathcal{L}$ in (8), we clearly have $\mathcal{L}(\bar{\mathbf{x}}^K, \boldsymbol{\theta}^*, \boldsymbol{\lambda}^*) - \mathcal{L}(\mathbf{x}^*, \boldsymbol{\theta}^*, \boldsymbol{\lambda}^*) \geq 0$; therefore,

$$\Phi(\bar{\mathbf{x}}^K) - \Phi(\mathbf{x}^*) + \langle \boldsymbol{\lambda}^*, M\bar{\mathbf{x}}^K \rangle + \sum_{i \in \mathcal{N}} \langle \theta_i^*, A_i\bar{x}_i^K - b_i \rangle \geq 0. \tag{38}$$

Recall that $i \in \mathcal{N}$ we defined $\tilde{w}_i^1 = \mathcal{P}_{\mathcal{K}_i}(\tilde{w}_i)$ and $\tilde{w}_i^2 = \mathcal{P}_{\mathcal{K}_i^\circ}(\tilde{w}_i)$, where $\tilde{w}_i \triangleq A_i\bar{x}_i^K - b_i \in \mathbb{R}^{m_i}$. For all $i \in \mathcal{N}$, $\theta_i^* \in \mathcal{K}_i^\circ$ and $\tilde{w}_i^1 \in \mathcal{K}_i$ imply $\langle \theta_i^*, \tilde{w}_i^1 \rangle \leq 0$; hence, for all $i \in \mathcal{N}$,

$$\langle A_i\bar{x}_i^K - b_i, \theta_i^* \rangle = \langle \tilde{w}_i - \tilde{w}_i^1 + \tilde{w}_i^1, \theta_i^* \rangle \leq \langle \tilde{w}_i - \tilde{w}_i^1, \theta_i^* \rangle \leq \|\theta_i^*\|d_{\mathcal{K}_i}(A_i\bar{x}_i^K - b_i).$$

Together with (38), we conclude that

$$\Phi(\bar{\mathbf{x}}^K) - \Phi(\mathbf{x}^*) + \|\boldsymbol{\lambda}^*\|\|M\bar{\mathbf{x}}^K\| + \sum_{i \in \mathcal{N}} \|\theta_i^*\|d_{\mathcal{K}_i}(A_i\bar{x}_i^K - b_i) \geq 0. \tag{39}$$

By combining inequalities (37) and (39) immediately implies the desired result. Finally, since (10) in Lemma 2.1 holds with strict inequality, $\bar{\mathbf{Q}}(A, A_0) \succ 0$. Hence, the proof of convergence for $\{\mathbf{x}^k\}$ and $\{\bar{\mathbf{x}}^k\}$ follows from Theorem 1.1.

## 5.4 Proof of Theorem 3.2

In order to prove Theorem 3.2, we first prove Theorem 5.2 which help us to appropriately bound $\mathcal{L}(\bar{\mathbf{x}}^K, \mathbf{y}) - \mathcal{L}(\mathbf{x}, \bar{\mathbf{y}}^K)$. Next, we provide a technical result in Lemma 5.3 to study the error accumulation, and another technical result in Lemma 5.4 to show the asymptotic convergence of $\{\mathbf{x}^k, \mathbf{y}^k\}$.

**Theorem 5.2.** *Let* $\mathbf{y} = [\boldsymbol{\theta}^\top \boldsymbol{\mu}^\top]^\top$ *such that* $\boldsymbol{\mu} \in \mathbb{R}^{n|\mathcal{N}|}$, $\boldsymbol{\theta} = [\theta_i]_{i \in \mathcal{N}} \in \mathbb{R}^m$, *and* $m \triangleq \sum_{i \in \mathcal{N}} m_i$; *and* $\{\mathbf{x}^k, \mathbf{y}^k\}_{k \geq 0}$ *be the iterate sequence generated using Algorithm DPDA-D, displayed in Fig. 2, initialized from an arbitrary* $\mathbf{x}^0$ *and* $\mathbf{y}^0$; *and* $\{\mathbf{e}^k\}_{k \geq 1}$ *be the proximal error sequence defined as in (20). For any* $\mathbf{x} \in \mathcal{X}$, *and* $\mathbf{y} \in \mathcal{Y}$, *the iterate sequence* $\{\mathbf{x}^k, \mathbf{y}^k\}_{k \geq 0}$ *satisfies for all* $k \geq 0$

$$\mathcal{L}(\mathbf{x}^{k+1}, \mathbf{y}) - \mathcal{L}(\mathbf{x}, \mathbf{y}^{k+1}) \leq E^{k+1}(\boldsymbol{\mu}) + \left[D_x(\mathbf{x}, \mathbf{x}^k) + D_y(\mathbf{y}, \mathbf{y}^k) - \langle T(\mathbf{x} - \mathbf{x}^k), \mathbf{y} - \mathbf{y}^k \rangle\right] \tag{40}$$

$$- \left[D_x(\mathbf{x}, \mathbf{x}^{k+1}) + D_y(\mathbf{y}, \mathbf{y}^{k+1}) - \langle T(\mathbf{x} - \mathbf{x}^{k+1}), \mathbf{y} - \mathbf{y}^{k+1} \rangle\right] - \frac{1}{2}(\mathbf{z}^{k+1} - \mathbf{z}^k)^\top \bar{\mathbf{Q}}(\mathbf{z}^{k+1} - \mathbf{z}^k),$$

*where* $\mathbf{z}^k = [\mathbf{x}^{k\top} \mathbf{y}^{k\top}]^\top$, $D_x$, $D_y$ *are Bregman functions defined as in Definition 1,* $T = [A^\top A_0^\top]^\top$ *for block-diagonal matrix* $A \triangleq \mathbf{diag}([A_i]_{i \in \mathcal{N}}) \in \mathbb{R}^{m \times n|\mathcal{N}|}$ *and* $A_0 = \mathbf{I}_{n|\mathcal{N}|}$, $\bar{\mathbf{Q}} \triangleq \bar{\mathbf{Q}}(A, A_0)$ *is defined as in Theorem 1.1 for* $A_0 = \mathbf{I}_{n|\mathcal{N}|}$, *and* $E^k(\boldsymbol{\mu}) \triangleq \|\mathbf{e}^k\|(2\gamma\sqrt{N}B + \|\boldsymbol{\mu} - \boldsymbol{\mu}^k\|)$ *for* $k \geq 1$.

*Proof.* For $k \geq 0$, let $\mathbf{q}^k \triangleq 2\mathbf{x}^{k+1} - \mathbf{x}^k$. From strong convexity of $\sigma_C(\boldsymbol{\mu}) - \langle \mathbf{q}^k, \boldsymbol{\mu} \rangle + \frac{1}{2\gamma} \|\boldsymbol{\mu} - \boldsymbol{\mu}^k\|_2^2$ in $\boldsymbol{\mu}$ and the fact that $\boldsymbol{\lambda}^{k+1}$ is its minimizer we conclude that

$$\sigma_C(\boldsymbol{\mu}) - \langle \mathbf{q}^k, \boldsymbol{\mu} \rangle + \frac{1}{2\gamma} \|\boldsymbol{\mu} - \boldsymbol{\mu}^k\|^2 \geq \sigma_C(\boldsymbol{\lambda}^{k+1}) - \langle \mathbf{q}^k, \boldsymbol{\lambda}^{k+1} \rangle + \frac{1}{2\gamma} \|\boldsymbol{\lambda}^{k+1} - \boldsymbol{\mu}^k\|^2 + \frac{1}{2\gamma} \|\boldsymbol{\mu} - \boldsymbol{\lambda}^{k+1}\|^2.$$

According to (20), $\boldsymbol{\mu}^k = \boldsymbol{\lambda}^k + \gamma \mathbf{e}^k$ for all $k \geq 1$; hence, from (22) we have

$$\sigma_C(\boldsymbol{\mu}) - \langle \mathbf{q}^k, \boldsymbol{\mu} \rangle + \frac{1}{2\gamma} \|\boldsymbol{\mu} - \boldsymbol{\mu}^k\|_2^2 \geq$$
$$\sigma_C(\boldsymbol{\mu}^{k+1}) - \langle \mathbf{q}^k, \boldsymbol{\mu}^{k+1} \rangle + \frac{1}{2\gamma} \|\boldsymbol{\mu}^{k+1} - \boldsymbol{\mu}^k\|_2^2 + \frac{1}{2\gamma} \|\boldsymbol{\mu} - \boldsymbol{\mu}^{k+1}\|_2^2 - S^{k+1}(\boldsymbol{\mu}), \qquad (41)$$

where the error term $S^{k+1}(\boldsymbol{\mu})$ is defined as

$$S^{k+1}(\boldsymbol{\mu}) \triangleq \gamma \sqrt{N} B \|\mathbf{e}^{k+1}\| - \gamma \|\mathbf{e}^{k+1}\|^2 - \left\langle \mathbf{e}^{k+1}, \boldsymbol{\mu} - 2\boldsymbol{\mu}^{k+1} + \boldsymbol{\mu}^k + \gamma \mathbf{q}^k \right\rangle. \qquad (42)$$

If one customizes the steps of Lemma 5.1 for problem (11) using $\boldsymbol{\mu}^{k+1}$ instead of $\boldsymbol{\lambda}^{k+1}$, it immediately follows from (41) that for all $k \geq 0$:

$$\mathcal{L}(\mathbf{x}^{k+1}, \mathbf{y}) - \mathcal{L}(\mathbf{x}, \mathbf{y}^{k+1}) \leq S^{k+1}(\boldsymbol{\mu}) + \left[ D_x(\mathbf{x}, \mathbf{x}^k) + D_y(\mathbf{y}, \mathbf{y}^k) - \left\langle T(\mathbf{x} - \mathbf{x}^k), \mathbf{y} - \mathbf{y}^k \right\rangle \right] \qquad (43)$$
$$- \left[ D_x(\mathbf{x}, \mathbf{x}^{k+1}) + D_y(\mathbf{y}, \mathbf{y}^{k+1}) - \left\langle T(\mathbf{x} - \mathbf{x}^{k+1}), \mathbf{y} - \mathbf{y}^{k+1} \right\rangle \right] - \frac{1}{2} (\mathbf{z}^{k+1} - \mathbf{z}^k)^\top \bar{\mathbf{Q}} (\mathbf{z}^{k+1} - \mathbf{z}^k),$$

where $\mathbf{z}^k = [\mathbf{x}^{k\top} \mathbf{y}^{k\top}]^\top$, and $\bar{\mathbf{Q}} \triangleq \bar{\mathbf{Q}}(A, A_0)$ is defined as in Theorem 1.1 for $A_0 = \mathbf{I}_{n|\mathcal{N}|}$.

For $k \geq 0$, let $\mathbf{h}^{k+1} \triangleq \mathcal{P}_C(\frac{1}{\gamma}\boldsymbol{\mu}^k + \mathbf{q}^k)$; hence, $\boldsymbol{\lambda}^{k+1} = \boldsymbol{\mu}^k + \gamma \mathbf{q}^k - \gamma \mathbf{h}^{k+1}$. Since $\boldsymbol{\mu}^{k+1} = \boldsymbol{\lambda}^{k+1} + \gamma \mathbf{e}^{k+1}$, we have $\boldsymbol{\mu}^k + \gamma \mathbf{q}^k - \boldsymbol{\mu}^{k+1} = \gamma(\mathbf{h}^{k+1} - \mathbf{e}^{k+1})$; therefore, (42) can be written as

$$S^{k+1}(\boldsymbol{\mu}) = \gamma \sqrt{N} B \|\mathbf{e}^{k+1}\| - \left\langle \mathbf{e}^{k+1}, \boldsymbol{\mu} - \boldsymbol{\mu}^{k+1} + \gamma \mathbf{h}^k \right\rangle \leq E^{k+1}(\boldsymbol{\mu}), \qquad (44)$$

where the inequality follows from Cauchy-Schwarz inequality and the fact that $\|\mathbf{h}^{k+1}\| \leq \sqrt{N} B$ since $\mathbf{h}^{k+1} \in C$. Combining (43) and (44) gives the desired result. $\qquad \square$

The Lemma 5.3 below is a slight extension of Proposition 3 in [20], where it is stated for $q = 1$. The proof is omitted due to limited space. The next result in Lemma 5.4 follows from [23].

**Lemma 5.3.** *Let* $\alpha \in (0,1)$, $q \geq 1$ *is a rational number, and* $d \in \mathbb{Z}_+$. *Define* $P(k, d) = \{\sum_{i=0}^d c_i k^i : c_i \in \mathbb{R} \ i = 1, \dots, d\}$ *denote the set of polynomials of* $k$ *with degree at most* $d$. *Suppose* $p^{(k)} \in P(k, d)$ *for* $k \geq 1$, *then* $\sum_{k=0}^{+\infty} p^{(k)} \alpha^{\sqrt[q]{k}}$ *is finite.*

**Lemma 5.4.** *Let* $\{a^k\}$, $\{b^k\}$, *and* $\{c^k\}$ *be non-negative real sequences such that* $a^{k+1} \leq a^k - b^k + c^k$ *for all* $k \geq 1$, *and* $\sum_{k=1}^\infty c^k < \infty$. *Then* $a = \lim_{k \to \infty} a^k$ *exists, and* $\sum_{k=1}^\infty b^k < \infty$.

Now we are ready to prove Theorem 3.2.

## 5.5 Proof of Theorem 3.2

Setting $A_0 = \mathbf{I}_{n|\mathcal{N}|}$ instead of $M$ in the proof of Lemma 2.1, one can show that $\bar{\mathbf{Q}} = \bar{\mathbf{Q}}(A, A_0) \succ 0$ when the condition in (23) holds for all $i \in \mathcal{N}$; thus, we can drop the last term in (40). Similar to the proof of Theorem 1.1, summing (40) over $k$ after dropping $-\frac{1}{2} \|\mathbf{z}^{k+1} - \mathbf{z}^k\|_{\bar{\mathbf{Q}}}$, using Jensen's inequality, and dropping the last term, $D_x(\mathbf{x}, \mathbf{x}^K) + D_y(\mathbf{y}, \mathbf{y}^K) - \langle T(\mathbf{x} - \mathbf{x}^K), \mathbf{y} - \mathbf{y}^K \rangle \geq 0$, in the telescoping sum gives

$$\mathcal{L}(\bar{\mathbf{x}}^K, \mathbf{y}) - \mathcal{L}(\mathbf{x}, \bar{\mathbf{y}}^K) \leq \frac{1}{K} \left[ D_x(\mathbf{x}, \mathbf{x}^0) + D_y(\mathbf{y}, \mathbf{y}^0) - \langle T(\mathbf{x} - \mathbf{x}^0), \mathbf{y} - \mathbf{y}^0 \rangle + \sum_{k=1}^K E^k(\boldsymbol{\mu}) \right]. \qquad (45)$$

Note that $E^k(\boldsymbol{\mu})$ appearing in (45) is the error term due to approximating $\mathcal{P}_C$ in the $k$-th iteration of the algorithm for $k \geq 1$. Furthermore, (45) can be written more explicitly as follows: let $\bar{\mathbf{x}}^K \triangleq \frac{1}{K} \sum_{k=1}^K \mathbf{x}^k$, $\bar{\boldsymbol{\mu}}^K \triangleq \frac{1}{K} \sum_{k=1}^K \boldsymbol{\mu}^k$, and $\bar{\boldsymbol{\theta}}^K \triangleq \frac{1}{K} \sum_{k=1}^K \boldsymbol{\theta}^k$, then for any $\mathbf{x}, \boldsymbol{\mu} \in \mathbb{R}^{n|\mathcal{N}|}, \boldsymbol{\theta} \in \mathbb{R}^m$

such that $m = \sum_{i \in \mathcal{N}} m_i$, and for all $K \geq 1$, we have

$$\mathcal{L}(\bar{\mathbf{x}}^K, \boldsymbol{\theta}, \boldsymbol{\mu}) - \mathcal{L}(\mathbf{x}, \bar{\boldsymbol{\theta}}^K, \bar{\boldsymbol{\mu}}^K) \leq \Theta(\mathbf{x}, \boldsymbol{\theta}, \boldsymbol{\mu})/K,$$

$$\Theta(\mathbf{x}, \boldsymbol{\theta}, \boldsymbol{\mu}) \triangleq \frac{1}{2\gamma}\|\boldsymbol{\mu} - \boldsymbol{\mu}^0\|^2 - \langle \mathbf{x} - \mathbf{x}^0, \ \boldsymbol{\mu} - \boldsymbol{\mu}^0 \rangle + \sum_{k=1}^{K} E^k(\boldsymbol{\mu})$$

$$+ \sum_{i \in \mathcal{N}} \left[ \frac{1}{2\tau_i}\|x_i - x_i^0\|^2 + \frac{1}{2\kappa_i}\|\theta_i - \theta_i^0\|^2 - \langle A_i(x_i - x_i^0), \theta_i - \theta_i^0 \rangle \right].$$

Note that under the assumption in (23), Schur complement condition guarantees that

$$\begin{bmatrix} \frac{1}{\tau_i}\mathbf{I}_n & -A_i^\top \\ -A_i & \frac{1}{\kappa_i}\mathbf{I}_{m_i} \end{bmatrix} \preceq \begin{bmatrix} \frac{2}{\tau_i}\mathbf{I}_n & \mathbf{0}^\top \\ \mathbf{0} & \frac{2}{\kappa_i}\mathbf{I}_{m_i} \end{bmatrix}.$$

Therefore,

$$\Theta(\mathbf{x}, \boldsymbol{\theta}, \boldsymbol{\mu}) \leq \sum_{i \in \mathcal{N}} \left[ \frac{1}{\tau_i}\|x_i - x_i^0\|^2 + \frac{1}{\kappa_i}\|\theta_i - \theta_i^0\|^2 \right] + \frac{1}{2\gamma}\|\boldsymbol{\mu} - \boldsymbol{\mu}^0\|^2$$

$$- \langle \mathbf{x} - \mathbf{x}^0, \boldsymbol{\mu} - \boldsymbol{\mu}^0 \rangle + \sum_{k=1}^{K} E^k(\boldsymbol{\mu}). \tag{46}$$

As argued in the proof of Theorem 2.2, if Assumption 1.1 holds, one can construct a saddle point $(\mathbf{x}^*, \boldsymbol{\theta}^*, \boldsymbol{\lambda}^*)$ for $\mathcal{L}$ in (11); hence, $\mathcal{L}(\mathbf{x}^*, \boldsymbol{\theta}^*, \boldsymbol{\lambda}^*) = \Phi(\mathbf{x}^*)$ and $\theta_i^* \in \mathcal{K}_i^\circ$ for $i \in \mathcal{N}$. As in the proof of Theorem 2.2, define $\tilde{\boldsymbol{\theta}} = [\tilde{\theta}_i]_{i \in \mathcal{N}}$ such that $\tilde{\theta}_i \triangleq 2\|\theta_i^*\| \left( \|\mathcal{P}_{\mathcal{K}_i^\circ}(A_i \bar{x}_i^K - b_i)\| \right)^{-1} \mathcal{P}_{\mathcal{K}_i^\circ}(A_i \bar{x}_i^K - b_i) \in \mathcal{K}_i^\circ$, which implies

$$\langle A_i \bar{x}_i^K - b_i, \tilde{\theta}_i \rangle = 2\|\theta_i^*\| d_{\mathcal{K}_i}(A_i \bar{x}_i^K - b_i). \tag{47}$$

Define $\tilde{C} \triangleq \{\mathbf{x} \in \mathbb{R}^{n|\mathcal{N}|} : \exists \bar{x} \in \mathbb{R}^n \text{ s.t. } x_i = \bar{x}, \forall i \in \mathcal{N}\}$. Note that $\tilde{C}$ is a closed convex cone, and the projection $\mathcal{P}_{\tilde{C}}(\mathbf{x}) = \mathbf{1} \otimes \tilde{p}(\mathbf{x})$, where $\tilde{p}(\mathbf{x})$ is defined in (15). Let $\tilde{\boldsymbol{\mu}} = 2\|\boldsymbol{\lambda}^*\| \frac{\mathcal{P}_{\tilde{C}^\circ}(\bar{\mathbf{x}}^K)}{\|\mathcal{P}_{\tilde{C}^\circ}(\bar{\mathbf{x}}^K)\|} \in \tilde{C}^\circ$, where $\tilde{C}^\circ$ denotes polar cone of $\tilde{C}$. Hence, it can be verified that $\langle \tilde{\boldsymbol{\mu}}, \bar{\mathbf{x}}^K \rangle = 2\|\boldsymbol{\lambda}^*\| d_{\tilde{C}}(\bar{\mathbf{x}}^K)$. Note that $\tilde{\boldsymbol{\mu}} \in \tilde{C}^\circ$ implies that $\sigma_{\tilde{C}}(\tilde{\boldsymbol{\mu}}) = 0$; moreover, we also have $C \subseteq \tilde{C}$; hence, $\sigma_C(\tilde{\boldsymbol{\mu}}) \leq \sigma_{\tilde{C}}(\tilde{\boldsymbol{\mu}}) = 0$. Therefore, we can conclude that $\sigma_C(\tilde{\boldsymbol{\mu}}) = 0$ since $\mathbf{0} \in C$. Together with (47), we get

$$\mathcal{L}(\bar{\mathbf{x}}^K, \tilde{\boldsymbol{\theta}}, \tilde{\boldsymbol{\mu}}) - \mathcal{L}(\mathbf{x}^*, \boldsymbol{\theta}^*, \boldsymbol{\lambda}^*) = \Phi(\bar{\mathbf{x}}^K) - \Phi(\mathbf{x}^*) + 2 \left( \|\boldsymbol{\lambda}^*\| d_{\tilde{C}}(\bar{\mathbf{x}}^K) + \sum_{i \in \mathcal{N}} d_{\mathcal{K}_i}(A_i \bar{x}_i^K - b_i) \|\theta_i^*\| \right). \tag{48}$$

Now we are going to upper bound $\Theta(\mathbf{x}^*, \tilde{\boldsymbol{\theta}}, \tilde{\boldsymbol{\mu}})$ using (46). Since $\boldsymbol{\mu}^0 = \mathbf{0}$, from Cauchy-Schwarz inequality,

$$|\langle \mathbf{x}^* - \mathbf{x}^0, \tilde{\boldsymbol{\mu}} - \boldsymbol{\mu}^0 \rangle| \leq 2\|\boldsymbol{\lambda}^*\| \|\mathbf{x}^* - \mathbf{x}^0\|. \tag{49}$$

Since $\boldsymbol{\theta}^*$ and $\boldsymbol{\lambda}^*$ maximize the Lagrangian function at $\mathbf{x}^*$, and $\boldsymbol{\theta}^0 = \mathbf{0}$, it follows from (47), (49), and (46) that

$$\mathcal{L}(\bar{\mathbf{x}}^K, \tilde{\boldsymbol{\theta}}, \tilde{\boldsymbol{\mu}}) - \mathcal{L}(\mathbf{x}^*, \boldsymbol{\theta}^*, \boldsymbol{\lambda}^*) \leq \mathcal{L}(\bar{\mathbf{x}}^K, \tilde{\boldsymbol{\theta}}, \tilde{\boldsymbol{\mu}}) - \mathcal{L}(\mathbf{x}^*, \bar{\boldsymbol{\theta}}^K, \bar{\boldsymbol{\mu}}^K)$$

$$\leq \frac{1}{K}\Theta(\mathbf{x}^*, \tilde{\boldsymbol{\theta}}, \tilde{\boldsymbol{\mu}}) \leq \frac{1}{K}\left( \Theta_2 + \sum_{k=1}^{K} E^k(\tilde{\boldsymbol{\mu}}) \right).$$

Therefore, we can conclude that

$$\Phi(\bar{\mathbf{x}}^K) - \Phi(\mathbf{x}^*) + 2\left( \|\boldsymbol{\lambda}^*\| d_{\tilde{C}}(\bar{\mathbf{x}}^K) + \sum_{i \in \mathcal{N}} d_{\mathcal{K}_i}(A_i \bar{x}_i^K - b_i) \|\theta_i^*\| \right) \leq \frac{1}{K}\left( \Theta_2 + \sum_{k=1}^{K} E^k(\tilde{\boldsymbol{\mu}}) \right), \tag{50}$$

where we use $\mathcal{L}(\mathbf{x}^*, \boldsymbol{\theta}^*, \boldsymbol{\lambda}^*) = \Phi(\mathbf{x}^*)$ and the fact that $\sigma_{\mathcal{K}_i}(\tilde{\theta}_i) = 0$ due to (32) since $\tilde{\theta}_i \in \mathcal{K}_i^\circ$.

Since $(\mathbf{x}^*, \boldsymbol{\theta}^*, \boldsymbol{\lambda}^*)$ is a saddle-point for $\mathcal{L}$ in (11), we clearly have $\mathcal{L}(\bar{\mathbf{x}}^K, \boldsymbol{\theta}^*, \boldsymbol{\lambda}^*) - \mathcal{L}(\mathbf{x}^*, \boldsymbol{\theta}^*, \boldsymbol{\lambda}^*) \geq 0$; therefore,

$$\Phi(\bar{\mathbf{x}}^K) - \Phi(\mathbf{x}^*) + \langle \boldsymbol{\lambda}^*, \bar{\mathbf{x}}^K \rangle + \sum_{i \in \mathcal{N}} \langle \theta_i^*, A_i \bar{\mathbf{x}}_i^K - b_i \rangle \geq 0. \tag{51}$$

As shown in the proof of Theorem 2.2, for all $i \in \mathcal{N}$, we have

$$\langle A_i \bar{\mathbf{x}}_i^K - b_i, \theta_i^* \rangle \leq \|\theta_i^*\| d_{\mathcal{K}_i}(A_i \bar{\mathbf{x}}_i^K - b_i).$$

Similarly, we can also show that $\langle \boldsymbol{\lambda}^*, \ \bar{\mathbf{x}}^K \rangle \leq \|\boldsymbol{\lambda}^*\| d_{\tilde{C}}(\bar{\mathbf{x}}^K)$. Together with (51), we conclude that

$$\Phi(\bar{\mathbf{x}}^K) - \Phi(\mathbf{x}^*) + \|\boldsymbol{\lambda}^*\| d_{\tilde{C}}(\bar{\mathbf{x}}^K) + \sum_{i \in \mathcal{N}} \|\theta_i^*\| d_{\mathcal{K}_i}(A_i \bar{\mathbf{x}}_i^K - b_i) \geq 0. \tag{52}$$

Finally, note that (18) implies that $\|\mathcal{R}^k(\mathbf{x}) - \mathcal{P}_C(\mathbf{x})\| \leq N\, \Gamma \alpha^{q_k} \|\mathbf{x}\|$ for all $\mathbf{x}$, and $\|\mathbf{x}^k\| \leq \sqrt{N}\, B$ for $k \geq 1$, it follows from (20) and (21) that

$$\|\mathbf{e}^{k+1}\| = \|\mathcal{P}_C\left(\tfrac{1}{\gamma}\boldsymbol{\mu}^k + 2\mathbf{x}^{k+1} - \mathbf{x}^k\right) - \mathcal{R}^k\left(\tfrac{1}{\gamma}\boldsymbol{\mu}^k + 2\mathbf{x}^{k+1} - \mathbf{x}^k\right)\|$$

$$\leq N\, \Gamma \alpha^{q_k} \left\|\tfrac{1}{\gamma}\boldsymbol{\mu}^k + 2\mathbf{x}^{k+1} - \mathbf{x}^k\right\| \leq 4N^{\frac{3}{2}} B \Gamma \alpha^{q_k}(k+1). \tag{53}$$

Moreover, using (53) and (21) we obtain that

$$\sum_{k=1}^K E^k(\tilde{\boldsymbol{\mu}}) = \sum_{k=1}^K \|\mathbf{e}^k\| \left(2\gamma\sqrt{N}\, B + \|\tilde{\boldsymbol{\mu}} - \boldsymbol{\mu}^k\|\right) \leq \sum_{k=1}^K 4N^{\frac{3}{2}} B \Gamma \alpha^{q_k} k \left(2\gamma\sqrt{N}\, B + \|\tilde{\boldsymbol{\mu}} - \boldsymbol{\mu}^k\|\right)$$

$$\leq 8N^2 B^2 \Gamma \sum_{k=1}^K \alpha^{q_k} \left[2\gamma k^2 + \left(\gamma + \tfrac{\|\boldsymbol{\lambda}^*\|}{\sqrt{N}B}\right) k\right] = \Theta_3(K). \tag{54}$$

From Lemma 5.3, it follows that $\sup_{K \in \mathbb{Z}_+} \Theta_3(K) < \infty$. Therefore, combining inequalities (50), (52) and (54) immediately implies the desired result.

Let $\mathbf{z}^* = [\mathbf{x}^{*\top} \mathbf{y}^{*\top}]^\top$ be a saddle point for $\mathcal{L}$ in (11), where $\mathbf{y}^* = [\boldsymbol{\theta}^{*\top} \boldsymbol{\lambda}^{*\top}]^\top$. Due to (23), we have $\bar{\mathbf{Q}} \succ 0$; hence, evaluating (40) at $\mathbf{z} = \mathbf{z}^*$, we get $k \geq 0$

$$0 \leq \mathcal{L}(\mathbf{x}^{k+1}, \mathbf{y}^*) - \mathcal{L}(\mathbf{x}^*, \mathbf{y}^{k+1}) \leq E^{k+1}(\boldsymbol{\lambda}^*) + \left[D_x(\mathbf{x}^*, \mathbf{x}^k) + D_y(\mathbf{y}^*, \mathbf{y}^k) - \left\langle T(\mathbf{x}^* - \mathbf{x}^k), \ \mathbf{y}^* - \mathbf{y}^k \right\rangle\right]$$

$$- \left[D_x(\mathbf{x}^*, \mathbf{x}^{k+1}) + D_y(\mathbf{y}^*, \mathbf{y}^{k+1}) - \left\langle T(\mathbf{x}^* - \mathbf{x}^{k+1}), \ \mathbf{y}^* - \mathbf{y}^{k+1} \right\rangle\right]$$

$$- \tfrac{1}{2} \left\|\mathbf{z}^{k+1} - \mathbf{z}^k\right\|_{\bar{\mathbf{Q}}}^2. \tag{55}$$

Define $\mathbf{a}^k \triangleq D_x(\mathbf{x}^*, \mathbf{x}^k) + D_y(\mathbf{y}^*, \mathbf{y}^k) - \left\langle T(\mathbf{x}^* - \mathbf{x}^k), \ \mathbf{y}^* - \mathbf{y}^k \right\rangle$, $\mathbf{b}^k \triangleq \tfrac{1}{2} \left\|\mathbf{z}^{k+1} - \mathbf{z}^k\right\|_{\bar{\mathbf{Q}}}^2$, and $\mathbf{c}^k \triangleq E^{k+1}(\boldsymbol{\lambda}^*)$ for $k \geq 0$. Clearly, $\mathbf{b}^k \geq 0$ and $\mathbf{c}^k \geq 0$ for $k \geq 0$. Moreover, from the definition of $\bar{\mathbf{Q}}$, it follows that (28) holds for all $\mathbf{z}, \mathbf{z}'$; therefore, $\mathbf{a}^k \geq \tfrac{1}{2} \left\|\mathbf{z}^k - \mathbf{z}^*\right\|_{\mathbf{Q}}^2 \geq 0$ for $k \geq 0$. Finally, note that (54) also implies that $\sum_{k=1}^K E^k(\boldsymbol{\lambda}^*) \leq \Theta_3(K)$. Since $\sup_{K \in \mathbb{Z}_+} \Theta_3(K) < \infty$, Lemma 5.4 implies that $\lim_{k \to \infty} \mathbf{a}^k$ exists. Thus, $\{\mathbf{a}^k\}$ is a bounded sequence, and this also implies that $\{\mathbf{z}^k\}$ is bounded as well. Consequently, there exists a subsequence $\{\mathbf{z}^{k_n}\}_n$ such that $\mathbf{z}^{k_n} \to \hat{\mathbf{z}}$ as $n \to \infty$. Since (55) is true for any saddle point $\mathbf{z}^*$, by setting $\mathbf{z}^* = \hat{\mathbf{z}}$ in (55), one can conclude that $\mathbf{s} \triangleq \lim_k \mathbf{s}^k \geq 0$ exists, where

$$\mathbf{s}^k \triangleq D_x(\hat{\mathbf{x}}, \mathbf{x}^k) + D_y(\hat{\mathbf{y}}, \mathbf{y}^k) - \left\langle T(\hat{\mathbf{x}} - \mathbf{x}^k), \hat{\mathbf{y}} - \mathbf{y}^k \right\rangle, \tag{56}$$

for $k \geq 0$. Since $\mathbf{s} = \lim_n \mathbf{s}^{k_n}$ and $\lim_n \left\langle T(\hat{\mathbf{x}} - \mathbf{x}^{k_n}), \hat{\mathbf{y}} - \mathbf{y}^{k_n} \right\rangle = 0$ (from $\mathbf{z}^{k_n} \to \hat{\mathbf{z}}$), clearly

$$\mathbf{s} = \lim_{n \to \infty} D_x(\hat{\mathbf{x}}, \mathbf{x}^{k_n}) + D_y(\mathbf{y}^*, \mathbf{y}^{k_n}) = 0,$$

which also implies that $\mathbf{z}^k \to \hat{\mathbf{z}}$.

## 5.6 Additional figures

**Figure 4:** Comparison among DPDA-S on a random graph with algebraic connectivity $4$, local SVMs for two nodes, and central SVM against test data. All models are trained with $C = 2$.

**Figure 5:** Comparison among DPDA-S on a random graph with algebraic connectivity $4$, local SVMs for two nodes, and central SVM against training data. All models are trained with $C = 2$.