[Reviews · NeurIPS 2016]

Reviewer 1

Summary

In this work, the authors consider the problem of minimizing the sum of several function under linear, conic, constraints. The authors propose an Algorithm (DPDA-S) for solving this problem. The convergences of this algorithm is inverstigated. A dynamic version this work is also considered, the reusulting algorithm is DPDA-D. Numerical results of DPDA-S and DPDA-D is tesed on the primal-dual SVM.

Qualitative Assessment

The result on convergence of the iteration should be improved. The convergence of the dual sequence should be derive. More numrical results should be add. Some existing works are not cited.

Confidence in this Review

2-Confident (read it all; understood it all reasonably well)


Reviewer 2

Summary

Based on a centralized primal-dual method, the paper designs a distributed primal-dual method for minimizing a sum of potentially non-smooth convex objective functions with conic constraints, for both static network and dynamic network (via extra consensus steps). The authors analyzed the convergence and rate (for optimality, constraint violation and consensus gap). Simple numerical studies were also performed to verify the theoretical analysis.

Qualitative Assessment

This paper works on an important problem and is technically interesting. The writing, especially the inclusion of intuition and explanation, however, needs some improvement. The consensus step is hidden under the notation R and is not very obvious where it is in Algorithm 2. I'd recommend the authors leave the W^t matrix in the algorithm, or analyze the communication required explicitly, otherwise, it's hard for the readers to follow. Also a few spaces were missing: Section 1.1, "without any rate result.In" should read "without any rate result. In", Assumption 3.1, "infinitely manyt" should be "infinitely many t". Some suggestions to consider: 1. what if the compact domain assumption is missing, can we still provide guarantee? 2. For space constrained conference paper, instead of using lots of Kronecker product in R^n, it would be much simpler to consider the case in R and just say it can be extended 3. Numerical results against state of art algorithms would also be nice.

Confidence in this Review

3-Expert (read the paper in detail, know the area, quite certain of my opinion)


Reviewer 3

Summary

The papers minimizes the sum of agent-specific composite convex functions over agent-specific private conic constraints over an undirected static and time-varying networks. For the static network, this paper reformulates the problem into a form such that the primal-dual algorithm can apply. For the time-varying network, the difficulty is that an average is needed in each iteration and it is impossible or inefficient to get the average in a decentralized network in finite steps. So an approximation is applied.

Qualitative Assessment

This paper reformulate the decentralized problem into a form such that the primal-dual algorithm can apply. When the network is static, the algorithm is already decentralized, while for the time-varying network, an average is needed and an approximation is applied because an average will need too many loops of communication. The convergence rates in sub-optimality, infeasibility, and consensus are shown. The description in the numerical experiments needs more explanation.

Confidence in this Review

1-Less confident (might not have understood significant parts)


Reviewer 4

Summary

This paper develops a primal-dual algorithm for distributed optimization problems with private objective functions and (conic) constraints. When the network topology is static, the authors proposed an algorithm with O(1/K) convergence rate; when the topology is time varying, the authors proposed an approximate algorithm that offers similar guarantee as the static case.

Qualitative Assessment

The reviewer believes that the greatest contribution of this paper is the analysis done for the time varying case, where the authors transformed the problem into one that requires a projection step into the consensual set; which then allows for applying a multi-consensus steps, where an inexact update analysis for primal-dual algorithms (Thm. 5.2) can be applied. This results leads to a converging distributed PDA algorithm for time varying networks. That said, the reviewer has several concerns on the practicality of the results and the presentation of this paper. 1. To get around with the issue of computing (16)/(17) over time varying networks, the authors have adopted a multi-consensus step method to in (18) such that $q_k = k^{p/2}$ consensus steps are needed for each iteration $k$, i.e., the number of steps grows with iteration number. The reviewer wonders if it is a practical setup as this essentially requires the algorithm to be synchronous, i.e., every node starts iteration $k$ at the same time and knowing that he/she has to do exactly $q_k$ times of consensus. Also, as the network is time varying, the reviewer doubts if achieving such synchronization is possible since that will require sending a certain beacon signal that every agent can hear, or the nodes have to rely on some external communication protocol. 2. Besides some minor typos (which are gathered below as well), the reviewer finds the context of this paper difficult to grasp at times. Particularly, definitions for many of the constants / variables in the described algorithms are scattered around in the paper and some of the notations can be improved. E.g., "T" was not defined in Eq. (4), more intuitive descriptions should be given following each of the main results like Thm 2.2, 3.2, "s_i^{k}' in the DPDA-S algorithm can be defined similarly as $\bar{x}^k$, etc. Typos: 1. line 24, "our objective is to solve (3) in a decentralized..." 2. line 84, "it is shown ..." 3. line 156, the dimension for $\lambda$ should be $n |{\cal E}| x 1$. 4. line 211, it should be $N_i^t$ and in line 212, it should be $O_i^t$ as the network is time varying. 5. line 260, it should be $|S_{test}| = 600$. %%%%%%%%%%%%%% After authors' rebuttal %%%%%%%%%%%%%%%%%%%%%%%%%%%%%%% The reviewer is satisfied with the authors' response in overall. However, it is not clear to me on how to achieve synchronicity "by simply counting local communications with each neighbor", does that mean each node has to keep a certain local clock and count the number of local comm. rounds? As the network is time varying, it seems that some nodes may eventually miss a few round of communications if only the local clock is available. As such, I am retaining my previous rating as it seems that the current result is only useful in a time varying but synchronous network, which is not as interesting as the async. case as pointed out in my previous review.

Confidence in this Review

2-Confident (read it all; understood it all reasonably well)


Reviewer 5

Summary

The paper developed a fully distributed primal-dual algorithm by applying a general algorithm proposed by Chambolle and Pock to conic constrained consensus problem. The conic constraints give the new challenge in the consensus problem in addition to other settings of local communication. Both static and dynamic network topology settings are considered. For the static setting in Section 2, the algorithm is naturally distributed using the network topology, and the conic constraints are handled by projecting the dual variables of those constraints onto the polar cones. For dynamic/time-varying network in Section 3, the inexact consensus steps are needed to approximate the averaging operation in the \lambda update in a distributed way. The communication protocol in consensus step follows the work proposed by Nedic and Ozdaglar. It is shown that the algorithm converges at the 1/K rate.

Qualitative Assessment

The main contributions claimed by the authors include that the fully distributed algorithms are able to handle the conic constraints in the consensus problem. Indeed the algorithms proposed in the paper handle the conic constraints: project the dual variables corresponding to the conic constraints onto the polar cones. It is a better way to handle conic constraints with affine mapping than a more straightforward way which is to introduce additional variables and linear inequalities and project the additional variables onto the original cones. However, in the static setting this projection idea seems the only additional step contributed by the paper compared to a few existing primal-dual algorithms for unconstrained consensus problem, for instance, the proximal ADMM algorithm proposed in the paper "Proximal Alternating Direction Method of Multipliers for Distributed Optimization on Weighted Graphs" by Meng, Fazel and Mesbahi, CDC 2015. It would be necessary for authors to discuss the relation to more recent and active research on distributed primal-dual/proximal ADMM algorithms for consensus problems. Therefore, I think that the main novelty of the paper is on the dynamic/time-varying network topology where the difficulty of distributed operation comes from the projection onto the consensus set C. The projection onto C requires a global averaging step, so a multi-consensus step is used which leads to an inexact primal-dual algorithm. Compared to the overall clarity of the paper, it would be very necessary to extend numerical experiments: 1. the soft-margin SVM problem used as the example in this section can actually be formatted as an unconstrained consensus problem which can be solved by existing primal-dual algorithms. It would be necessary to compare the proposed algorithm for the constrained formulation to using the existing algorithms to unconstrained formulation; 2. The effect of algebraic connectivity is not successfully demonstrated in experiments: Figure 3 resembles Figure 6, so does Figure 4 to Figure 5. 3. All plots really should be plotted on the log scale to view the 1/K sublinear curve. This mathematical paper is very clearly presented and readable though the knowledge on the Chambolle and Pock algorithm would definitely help.

Confidence in this Review

3-Expert (read the paper in detail, know the area, quite certain of my opinion)


Reviewer 6

Summary

The authors consider the problem of decentralized optimization when each local objective is a composition of a convex smooth non-smooth term. They consider the saddle point formulation of this constrained optimization problem, and by considering its Lagrangian relaxation, propose a primal dual scheme to solve it. This method is particularly useful for cases where projection onto the local constraint set is easily computable. By deriving a closed form expression for the Lagrange multiplier update of this problem, an efficient algorithm is developed. It is shown that this method converge a primal dual optimal pair at a rate of $1/K$ when the primal vector is recursively averaged. Then the authors consider the extension of this method to a time-varying graph by incorporating an inner-loop communications protocol between neighboring agents, which leads to a modified dual update with several consensus steps per objective gradient update. The convergence guarantees of the static network case are generalized to the switching network case. Then, numerical analysis on a support vector classification problem demonstrate's the proposed methods empirical convergence behavior.

Qualitative Assessment

Generally speaking, there is not enough explanation of key quantities of interest in between the presentation of main results or ideas, e.g., at the beginning of Algorithm DPDA-S the intialization variables should also be explained (initialize primal variables, dual variables, learning rate, etc); The derivation of the fact that the dual variable update may be written in terms of the sequence $s_i^k$ requires more explanation (perhaps a display equation relating $s$ and $lambda$); Theorem 2.2 has no discussion...all theorems and lemmas need discussion/interpretation. The paper is not very readable as a result. The extension of the proposed method to time-varying topology could be omitted so that more explication and context for the algorithm derivation could fit. Nonetheless, the technical contributions of the paper are substantial in extending the primal-dual method to composite constrained convex programming in decentralized (time-varying network) settings. Numerical analysis for a decentralized support vector classification problem on a static network demonstrate the proposed method's practical utility, but are not very convincing for practitioners -- I find the multivariate Gaussian data example to be a mere toy problem. A switching network might arise in a cloud computing or computer network in which case a problem such as a playback buffer, resource allocation, or data packet routing would be more appropriate. --The bounded linear operator $T$ in equation (4) is left undefined, and had to be looked up in reference [11]. This makes the initial presentation of the proposed algorithm confusing, and not self-contained. Must $T$ be as in Theorem 1.1, or could it be any linear operator? --Also, why does a 2 appear in front of the inner-product term $\langle T( 2 x^{k+1} - x^k), y \rangle$ in equation (6)? This scaling does NOT appear in the update formula for (7) in reference [11]. This may be a mistake, but its carried through equations in the rest of the paper. -- In equation (8) the Lagrange multipliers $\lambda_{ij}$ are included in the optimization problem as a domain constraint but don't actually effect the objective or the constraint functions. Thus, this is probably a typo and should be deleted until the Lagrangian relaxation presented in (9).

Confidence in this Review

3-Expert (read the paper in detail, know the area, quite certain of my opinion)